# Advancing Pose-Guided Image Synthesis with Progressive Conditional Diffusion Models

**Fei Shen**,[*] **Hu Ye**[*], **Jun Zhang**[†], **Cong Wang, Xiao Han, Wei Yang**
Tencent AI Lab
{ffeishen, huye, junejzhang, xvencewang, haroldhan, willyang}@tencent.com

## Abstract

Recent work has showcased the significant potential of diffusion models in pose-guided person image synthesis. However, owing to the inconsistency in pose between the source and target images, synthesizing an image with a distinct pose, relying exclusively on the source image and target pose information, remains a formidable challenge. This paper presents **P**rogressive **C**onditional **D**iffusion **M**odel**s** (PCDMs) that incrementally bridge the gap between person images under the target and source poses through three stages. Specifically, in the first stage, we design a simple prior conditional diffusion model that predicts the global features of the target image by mining the global alignment relationship between pose coordinates and image appearance. Then, the second stage establishes a dense correspondence between the source and target images using the global features from the previous stage, and an inpainting conditional diffusion model is proposed to further align and enhance the contextual features, generating a coarse-grained person image. In the third stage, we propose a refining conditional diffusion model to utilize the coarsely generated image from the previous stage as a condition, achieving texture restoration and enhancing fine-detail consistency. The three-stage PCDMs work progressively to generate the final high-quality and high-fidelity synthesized image. Both qualitative and quantitative results demonstrate the consistency and photorealism of our proposed PCDMs under challenging scenarios. The code and model will be available at https://github.com/tencent-ailab/PCDMs.

## 1 Introduction

Given an image of a specific person under a particular pose, pose-guided image synthesis (Zhang et al., 2022; Ren et al., 2022; Bhunia et al., 2023) aims to generate images of the person with the same appearance and meanwhile under the given target pose, which more importantly, are expected to be as photorealistic as possible. It holds broad and robust application potential in e-commerce and content generation. Meanwhile, the generated images can be used to improve the performance of downstream tasks, such as person re-identification (Ye et al., 2021; Shen et al., 2023b). However, since pose disparities between the source and target images, generating an image with a different pose solely based on the source image and target pose information remains a significant challenge.

Previous work usually focuses on the generative adversarial network (GAN) (Creswell et al., 2018), variational autoencoder (VAE) (Kingma et al., 2019), and flow-based model (Li et al., 2019). GAN-base methods (Zhu et al., 2019; Tang et al., 2020) insert multiple repeating modules to mine the sparse correspondence between source and target pose image features. The outputs produced by these approaches often exhibit distorted textures, unrealistic body shapes, and localized blurriness, particularly when generating images of occluded body parts. In addition, owing to the nature of the adversarial min-max objective, GAN-based methods are susceptible to unstable training dynamics, limiting the diversity of the generated samples. Although VAE-based approaches (Siarohin et al., 2018; Esser et al., 2018) are relatively stable, they suffer from blurring of details and misalignment of target pose due to their reliance on surrogate loss for optimization. Flow-based methods (Li et al., 2019; Ren et al., 2021) have emerged to deal with this problem, which guide the source

---

[*]Equal contribution.
[†]Corresponding author

image features to distort to a reasonable target pose by predicting the correspondence between the source and target pose. However, when the source and target poses undergo large deformations or occlusions, this can easily lead to apparent artifacts in the generated images. Likewise, some methods (Lv et al., 2021; Zhang et al., 2021) utilize human parsing maps to learn the correspondence between image semantics and poses to ensure that the generated images are consistent with the target pose. Although these methods can generate images that meet pose consistency requirements, they still struggle to maintain consistent style and capture realistic texture details.

Recently, diffusion models (Bhunia et al., 2023; Zhang & Zhou, 2023) have made significant strides in the field of person image synthesis. They utilize the source image and target pose as conditions and generate the target image through a multi-step denoising process instead of completing it in a single step. So, these approaches help better retain the input information.

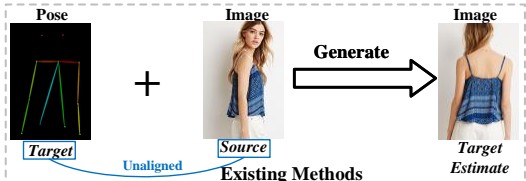

Figure 1: Existing methods typically utilize unaligned image-to-image generation at the conditional level.

tion. However, as shown in Figure 1 (a), due to the pose inconsistency between the source and target images, this essentially constitutes an unaligned image-to-image generation task at the conditional level. Moreover, the lack of dense correspondence between the source and target images regarding image, pose, and appearance often results in less realistic results.

This paper presents **P**rogressive **C**onditional **D**iffusion **M**odels (PCDMs) to tackle the aforementioned issues through three stages, as shown in Figure 1 (b). Initially, we propose a prior conditional diffusion model to predict global features given a target pose. This prediction is significantly simpler than directly generating target images, as we allow the prior model to concentrate solely on one task and thereby do not worry about realistically detailed texture generation. Given a source image and pose coordinates as conditions, the prior conditional diffusion model employs a transformer network to predict global features under the target pose. In the second stage, we use the global features from the previous stage to establish the dense correspondence between the source and target images, and propose an inpainting conditional diffusion model to further align and enhance contextual features, generating a coarse-grain synthetic image. Finally, we develop a refining conditional diffusion model. This model utilizes the coarse-grain image generated in the previous stage and applies post-image-to-image techniques to restore texture and enhance detail consistency. The three stages of the PCDMs operate progressively to generate visually more appealing outcomes, particularly when handling intricate poses.

We summarize the contributions of this paper as follows: (a) we devise a simple prior conditional diffusion model that explicitly generates and complements the embedding of the target image by mining the global alignment relationship between the source image appearance and the target pose coordinates. (b) we propose a novel inpainting conditional diffusion model to explore the dense correspondence between source and target images. (c) we introduce a new refining conditional diffusion model by further using post hoc image-to-image techniques to enhance the quality and fidelity of synthesized images. (d) We conduct comprehensive experiments on two public datasets to showcase the competitive performance of our method. Additionally, we implement a user study and a downstream task to evaluate the qualitative attributes of the images generated by our method.

## 2 RELATED WORK

**Person Image Synthesis.** The task of person image synthesis has achieved great development during these years, especially with the unprecedented success of deep learning. The earlier methods (Ma et al., 2017; Men et al., 2020) treat the synthesis task as conditional image generation, using conditional generative adversarial networks (CGANs) (Mirza & Osindero, 2014) to generate the target image with the source appearance image and target pose as conditions. However, due to the inconsistency between the source and target poses, the effectiveness of directly connecting the source image with the target pose is limited. To overcome this challenge, VUnet (Esser et al., 2018) adopts a joint application of VAE and U-Net to decouple the appearance and pose of the character image. Furthermore, Def-GAN (Siarohin et al., 2018) proposes a deformable GAN that decomposes the overall deformation through a set of local affine transformations to address the misalignment issues caused by different poses. On the other hand, some works (Liu et al., 2019; Ren et al., 2020) utilize flow-based deformation to transform source information, improving pose alignment. For example,

Figure 2: The three-stage pipeline of **P**rogressive **C**onditional **D**iffusion **M**odel**s** (PCDMs) progressively operates to generate the final high-quality and high-fidelity synthesized image. Our approach progressively predicts the global features, dense correspondences, and texture restoration of target image, enabling image synthesis.

GFLA (Ren et al., 2020) obtains global flow fields and occlusion masks to warp local patches of the source image to match the desired pose. Similarly, ClothFlow (Han et al., 2019) is a model designed based on this concept for segmentation-guided human image generation. Subsequently, methods such as PISE (Zhang et al., 2021), SPGnet (Lv et al., 2021), and CASD (Zhou et al., 2022) leverage parsing maps to generate the final image. CoCosNet (Zhou et al., 2021) extracts dense correspondence between cross-domain images through attention-based operations. However, person image synthesis is essentially a transformation from non-aligned images to images, and the absence of global appearance features of the target image can lead to less realistic results. Besides, there are some multi-stage methods in other generative domains. For example, Grigorev et al. (Grigorev et al., 2019) proposed a framework based on CNNs that first performs pose warping, followed by texture repair. Unselfie (LastName, 2014) introduces a pipeline that first identifies the target's neutral pose, repairs body texture, and then perfects and synthesizes the character in the background. While these methods can fit the target pose well, they lose a sense of realism when combined with the background or the human body.

**Diffusion Models.** Diffusion models (Ho et al., 2020; Song et al., 2020) have recently emerged as a prominent generative method, renowned for synthesizing high-quality images. Following the success in unconditional generation tasks, diffusion models have expanded to conditional generation tasks, demonstrating competitive and superior performance compared to GANs and VAEs. Unlike other generative methods, diffusion models employ a multi-step denoising process instead of generating the target image in a single step, which helps to better preserve input information. Moreover, this denoising process can enhance texture details, often producing sharper images than GANs and VAEs. Recent studies (Bhunia et al., 2023; Zhang & Zhou, 2023) have already explored person image synthesis based on diffusion models. MGD (Baldrati et al., 2023) guides the generation process by constraining a latent diffusion model with the model's pose, the garment sketch, and a textual description of the garment itself. PIDM (Bhunia et al., 2023) introduces a texture diffusion module and disentangled classifier-free guidance to ensure that the conditional input and the generated output are consistent regarding pose and appearance information. Given the robust generation capabilities of the diffusion model, we devise a framework with progressive conditional diffusion models, which consist of three pivotal stages: prior, inpainting, and refining.

## 3 METHOD

An overview of our **P**rogressive **C**onditional **D**iffusion **M**odel**s** (PCDMs) is described in Figure 2, which contains a prior conditional diffusion model, an inpainting conditional diffusion model, and a refining conditional diffusion model. Our method aims to leverage three-stage diffusion models to incrementally bridge the gap between person images under the target and source poses. The prior conditional diffusion model predicts the global features of the target image by mining the global alignment relationship between pose coordinates and image appearance (Section 3.2). Subsequently, the inpainting conditional diffusion model utilizes the global features from the previous stage to further enhance contextual features, generating a coarse-grained synthetic image (Section 3.3). Furthermore, the refining conditional diffusion model leverages the coarse-grained image generated in the prior stage, aiming to accomplish texture refinement and enhance detail consistency (Section 3.4).

### 3.1 PRELIMINARIES

**Diffusion Model.** Diffusion models are a type of generative models trained to reverse the diffusion process. The diffusion process gradually adds Gaussian noise to the data using a fixed Markov chain, while a denoising model is trained to generate samples from Gaussian noise. Given an input data

sample $x_0$ and an additional condition $c$, the training objective of diffusion model usually adopts a mean square error loss $L_{\text{simple}}$, as follows,

$$L_{\text{simple}} = \mathbb{E}_{\boldsymbol{x}_0, \boldsymbol{\epsilon} \sim \mathcal{N}(\mathbf{0}, \mathbf{I}), \boldsymbol{c}, t} \| \boldsymbol{\epsilon} - \boldsymbol{\epsilon}_\theta(\boldsymbol{x}_t, \boldsymbol{c}, t) \|^2, \tag{1}$$

where $\epsilon$ and $\epsilon_\theta$ represent the actual noise injected at the corresponding diffusion timestep $t$ and the noise estimated by the diffusion model $\theta$, respectively; $\boldsymbol{x}_t = \alpha_t \boldsymbol{x}_0 + \sigma_t \boldsymbol{\epsilon}$ is the noisy data at $t$ step, and $\alpha_t$, $\sigma_t$ are fixed functions of $t$ in the diffusion process. To reduce the computational resources, latent diffusion models (LDMs) (Rombach et al., 2022) operate the diffusion and denoising processes on the latent space encoded by a pretrained auto-encoder model.

**Classifier-Free Guidance.** In the context of conditional diffusion models, classifier-free guidance (Ho & Salimans, 2022) is a technique commonly used to balance image fidelity and sample diversity. During the training phase, conditional and unconditional diffusion models are jointly trained by randomly dropping $c$. In the sampling phase, the noise is predicted by the conditional model $\boldsymbol{\epsilon}_\theta(\boldsymbol{x}_t, \boldsymbol{c}, t)$ and the unconditional model $\boldsymbol{\epsilon}_\theta(\boldsymbol{x}_t, t)$ according to Eq. 2, as follows,

$$\hat{\boldsymbol{\epsilon}}_\theta(\boldsymbol{x}_t, \boldsymbol{c}, t) = w\boldsymbol{\epsilon}_\theta(\boldsymbol{x}_t, \boldsymbol{c}, t) + (1 - w)\boldsymbol{\epsilon}_\theta(\boldsymbol{x}_t, t), \tag{2}$$

where $w$ is the guidance scale used to control the strength of condition $c$.

## 3.2 PRIOR CONDITIONAL DIFFUSION MODEL

In the first stage, we propose a simple prior conditional diffusion model, designed to predict the global embedding of the target image. Here, we choose the image embedding extracted from CLIP (Radford et al., 2021) image encoder as the global embedding of the target image. CLIP is trained via contrastive learning on a large-scale image-text paired dataset. Hence, the image embedding can capture rich image content and style information, which can be used to guide subsequent target image synthesis.

As depicted in Figure 3, the prior conditional diffusion model is a transformer network, conditioned on the pose of the source image, the pose of the target image, and the source image. We first adopt OpenPose (Cao et al., 2017) to acquire the pose coordinates for the pose of source and target images. A compact trainable pose network, composed of 3 linear layers, is used to project the pose coordinates into the pose embedding. For the source image, we also use a CLIP image encoder to extract the image embedding and add a linear layer to project the image embedding. Additionally, we add an extra embedding to predict the unnoised global embedding of target image. The above embeddings plus timestep embedding and noisy image embedding of the target image are concatenated into a sequence of embeddings as the input of the transformer network.

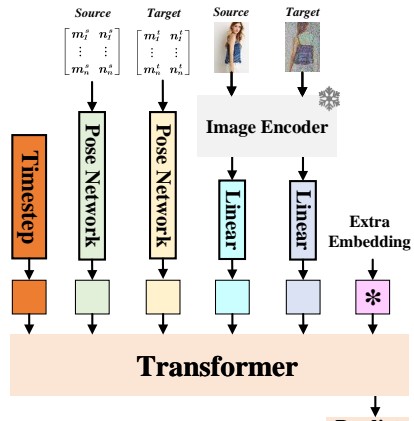

Figure 3: Illustration of the prior conditional diffusion model. The prior conditional diffusion model uses pose coordinates and global alignment relationship of the image to predict the global features of the target image.

Following unCLIP (Ramesh et al., 2022), the prior diffusion model is trained to predict the unnoised image embedding directly rather than the noise added to the image embedding. Given the source and target pose features $\boldsymbol{p}_s$ and $\boldsymbol{p}_t$, and the source image global feature $\boldsymbol{x}_s$, the training loss $L^{\text{prior}}$ of prior diffusion model $\boldsymbol{x}_\theta$ is defined as follows,

$$L^{\text{prior}} = \mathbb{E}_{\boldsymbol{x}_0, \boldsymbol{\epsilon}, \boldsymbol{x}_s, \boldsymbol{p}_s, \boldsymbol{p}_t, t} \| \boldsymbol{x}_0 - \boldsymbol{x}_\theta(\boldsymbol{x}_t, \boldsymbol{x}_s, \boldsymbol{p}_s, \boldsymbol{p}_t, t) \|^2. \tag{3}$$

Once the model learns the conditional distribution, the inference is performed according to Eq. 4, as follows,

$$\hat{\boldsymbol{x}}_\theta(\boldsymbol{x}_t, \boldsymbol{x}_s, \boldsymbol{p}_s, \boldsymbol{p}_t, t) = w\boldsymbol{x}_\theta(\boldsymbol{x}_t, \boldsymbol{x}_s, \boldsymbol{p}_s, \boldsymbol{p}_t, t) + (1 - w)\boldsymbol{x}_\theta(\boldsymbol{x}_t, t). \tag{4}$$

Figure 4: Overview of the inpainting conditional diffusion model. The inpainting conditional diffusion model utilizes the global features obtained from the previous stage to establish dense correspondences.

Figure 5: Illustrative of the refining conditional diffusion model. The refining conditional diffusion model leverages the coarse-grained image generated in the previous stage to rectify textures and ensure consistency.

### 3.3 INPAINTING CONDITIONAL DIFFUSION MODEL

With the global features of the target image obtained in the first stage, we propose an inpainting conditional diffusion model to establish dense correspondences between the source and target, and transform the unaligned image-to-image generation task into an aligned one. As shown in Figure 4, we concatenate the source and target images, the source and target poses, and the source and mask images along the width dimension. To prevent confusion between black and white in the source and target images, we add a single-channel marker symbol (omitted in the figure) with the same width and height as the input. We use 0 and 1 to represent masked and unmasked pixels, respectively. We then concatenate the global features of the target obtained from the prior conditional diffusion model (prior model) and the local features of the source image. This ensures that the input conditions of the model include the entirety of the source and target and are aligned at three levels: image, pose, and feature, which is overlooked in existing work.

Specifically, we use a pose encoder with four convolution layers similar to ControlNet (Zhang & Agrawala, 2023) to extract the pose features from the pose skeleton image. Unlike the prior model that uses pose coordinates, we expect this model to maintain image modality alignment throughout the learning phase, especially spatial information. For the source image, we use a frozen image encoder and a trainable MLP to extract the fine-grained features of the source image. Inspired by (Chen et al., 2023), we opt for DINOv2 (Oquab et al., 2023) as the image encoder because it can extract fine details. To better utilize the global features of the target image obtained from the previous stage, we also add it to the timestep embedding, which is embedded in the ResNet blocks of the entire network. The loss function $L^{\text{inpainting}}$ of inpainting conditional diffusion model according to Eq. 5, as follows,

$$L^{\text{inpainting}} = \mathbb{E}_{\boldsymbol{x}_0, \boldsymbol{\epsilon}, \boldsymbol{f}_{st}, \boldsymbol{p}_{st}, \boldsymbol{i}_{sm}, t} \| \boldsymbol{\epsilon} - \boldsymbol{\epsilon}_\theta(\boldsymbol{x}_t, \boldsymbol{f}_{st}, \boldsymbol{p}_{st}, \boldsymbol{i}_{sm}, t) \|^2, \tag{5}$$

where $\boldsymbol{f}_{st}$, $\boldsymbol{p}_{st}$, and $\boldsymbol{i}_{sm}$ respectively represent the feature embeddings obtained by concatenating the source and target global features, the feature embeddings of source and target poses, and the feature embeddings of source and mask images.

In the inference stage, we also use classifier-free guidance according to Eq. 6, as follows,

$$\hat{\boldsymbol{\epsilon}}_\theta(\boldsymbol{x}_t, \boldsymbol{f}_{st}, \boldsymbol{p}_{st}, \boldsymbol{i}_{sm}, t) = w\boldsymbol{\epsilon}_\theta(\boldsymbol{x}_t, \boldsymbol{f}_{st}, \boldsymbol{i}_{sm}, t) + (1 - w)\boldsymbol{\epsilon}_\theta(\boldsymbol{x}_t, \boldsymbol{p}_{st}, t). \tag{6}$$

### 3.4 REFINING CONDITIONAL DIFFUSION MODEL

Following the second stage, we obtain a preliminary generated coarse-grained target image. To further enhance the image quality and detail texture, as shown in Figure 5, we propose a refining conditional diffusion model. This model uses the coarse-grained image generated in the previous stage as a condition to improve the quality and fidelity of the synthesized image. We first concatenate the coarse-grained target image with the noisy image along the channel, which can be easily achieved by modifying the first convolutional layer of the diffusion model based on the UNet architecture. Then, we use the DINOv2 image encoder and a learnable MLP layer to extract features for the source image. Finally, we infuse texture features into the network through a cross-attention mechanism to guide the model in texture repair and enhance detail consistency.

Table 1: Quantitative comparison of the proposed PCDMs with several state-of-the-art models.

| Dataset | Methods | SSIM (↑) | LPIPS (↓) | FID (↓) |
|---|---|---|---|---|
| DeepFashion (Liu et al., 2016) (256 × 176) | Def-GAN (Siarohin et al., 2018) | 0.6786 | 0.2330 | 18.457 |
| | PATN (Zhu et al., 2019) | 0.6709 | 0.2562 | 20.751 |
| | ADGAN (Men et al., 2020) | 0.6721 | 0.2283 | 14.458 |
| | PISE (Zhang et al., 2021) | 0.6629 | 0.2059 | 13.610 |
| | GFLA (Ren et al., 2020) | 0.7074 | 0.2341 | 10.573 |
| | DPTN (Zhang et al., 2022) | 0.7112 | 0.1931 | 11.387 |
| | CASD (Zhou et al., 2022) | 0.7248 | 0.1936 | 11.373 |
| | NTED (Ren et al., 2022) | 0.7182 | 0.1752 | 8.6838 |
| | PIDM (Bhunia et al., 2023) | 0.7312 | 0.1678 | **6.3671** |
| | PCDMs *w/o* Refining | 0.7357 | 0.1426 | 7.7815 |
| | **PCDMs (Ours)** | **0.7444** | **0.1365** | 7.4734 |
| DeepFashion (Liu et al., 2016) (512 × 352) | CocosNet2 (Zhou et al., 2021) | 0.7236 | 0.2265 | 13.325 |
| | NTED (Ren et al., 2022) | 0.7376 | 0.1980 | 7.7821 |
| | PIDM (Bhunia et al., 2023) | 0.7419 | 0.1768 | **5.8365** |
| | PCDMs *w/o* Refining | 0.7532 | 0.1583 | 7.8422 |
| | **PCDMs (Ours)** | **0.7601** | **0.1475** | 7.5519 |
| Market-1501 (Zheng et al., 2015) (128 × 64) | Def-GAN (Siarohin et al., 2018) | 0.2683 | 0.2994 | 25.364 |
| | PTN (Zhu et al., 2019) | 0.2821 | 0.3196 | 22.657 |
| | GFLA (Ren et al., 2020) | 0.2883 | 0.2817 | 19.751 |
| | DPTN (Zhang et al., 2022) | 0.2854 | 0.2711 | 18.995 |
| | PIDM (Bhunia et al., 2023) | 0.3054 | 0.2415 | 14.451 |
| | PCDMs *w/o* Refining | 0.3107 | 0.2329 | 14.162 |
| | **PCDMs (Ours)** | **0.3169** | **0.2238** | **13.897** |

Assume that the coarse target features $i_{ct}$ and source image features $x_s$ are given, the loss function of refining conditional diffusion model is defined as follows,

$$L^{\text{refining}} = \mathbb{E}_{x_0, \epsilon, i_{ct}, x_s, t} \| \epsilon - \epsilon_\theta (x_t, i_{ct}, x_s, t) \|^2. \tag{7}$$

In the inference phase, we use the following Eq. 8,

$$\hat{\epsilon}_\theta(x_t, i_{ct}, f_s, t) = w\epsilon_\theta(x_t, i_{ct}, f_s, t) + (1 - w)\epsilon_\theta(x_t, t). \tag{8}$$

## 4 EXPERIMENTS

**Datasets.** We carry out experiments on DeepFashion (Liu et al., 2016), which consists of 52,712 high-resolution images of fashion models, and Market-1501 (Zheng et al., 2015) including 32,668 low-resolution images with diverse backgrounds, viewpoints, and lighting conditions. We extract the skeletons using OpenPose (Cao et al., 2017) and follow the dataset splits provided by (Bhunia et al., 2023). Note that the person ID of the training and testing sets do not overlap for both datasets.

**Metrics.** We conduct a comprehensive evaluation of the model, considering both objective and subjective metrics. Objective indicators include structural similarity index measure (SSIM) (Wang et al., 2004), learned perceptual image patch similarity (LPIPS) (Zhang et al., 2018), and fréchet inception distance (FID) (Heusel et al., 2017). In contrast, subjective assessments prioritize user-oriented metrics, including the percentage of real images misclassified as generated images (R2G) (Ma et al., 2017), the percentage of generated images misclassified as real images (G2R) (Ma et al., 2017), and the percentage of images deemed superior among all models (Jab) (Siarohin et al., 2018).

**Implementations.** We perform our experiments on 8 NVIDIA V100 GPUs. Our configurations can be summarized as follows: (1) the transformer of the prior model has 20 transformer blocks with a width of 2,048. For the inpainting model and refining model, we use the pretrained stable diffusion V2.1 [1] and modify the first convolution layer to adapt additional conditions. (2) We employ the AdamW optimizer with a fixed learning rate of $1e^{-4}$ in all stages. (3) Following (Ren et al., 2022; Bhunia et al., 2023), we train our models using images of sizes 256 × 176 and 512 × 352 for DeepFashion dataset. For the Market-1501 dataset, we utilize images of size 128 × 64. Please refer to B of the Appendix for more detail.

### 4.1 QUANTITATIVE AND QUALITATIVE RESULTS

We quantitatively compare our proposed PCDMs with several state-of-the-art methods, including Def-GAN (Siarohin et al., 2018), PATN (Zhu et al., 2019), ADGAN (Men et al., 2020), PISE (Zhang et al., 2021), GFLA (Ren et al., 2020), DPTN (Zhang et al., 2022), CASD (Zhou et al., 2022), CocosNet2 (Zhou et al., 2021), NTED (Ren et al., 2022) and PIDM (Bhunia et al., 2023).

**Quantitative Results.** From Table 1, PCDMs excels in two out of three metrics on the DeepFashion compared to other models, regardless of whether it is based on GAN, VAE, flow-based model, or

---

[1]https://huggingface.co/stabilityai/stable-diffusion-2-1-base

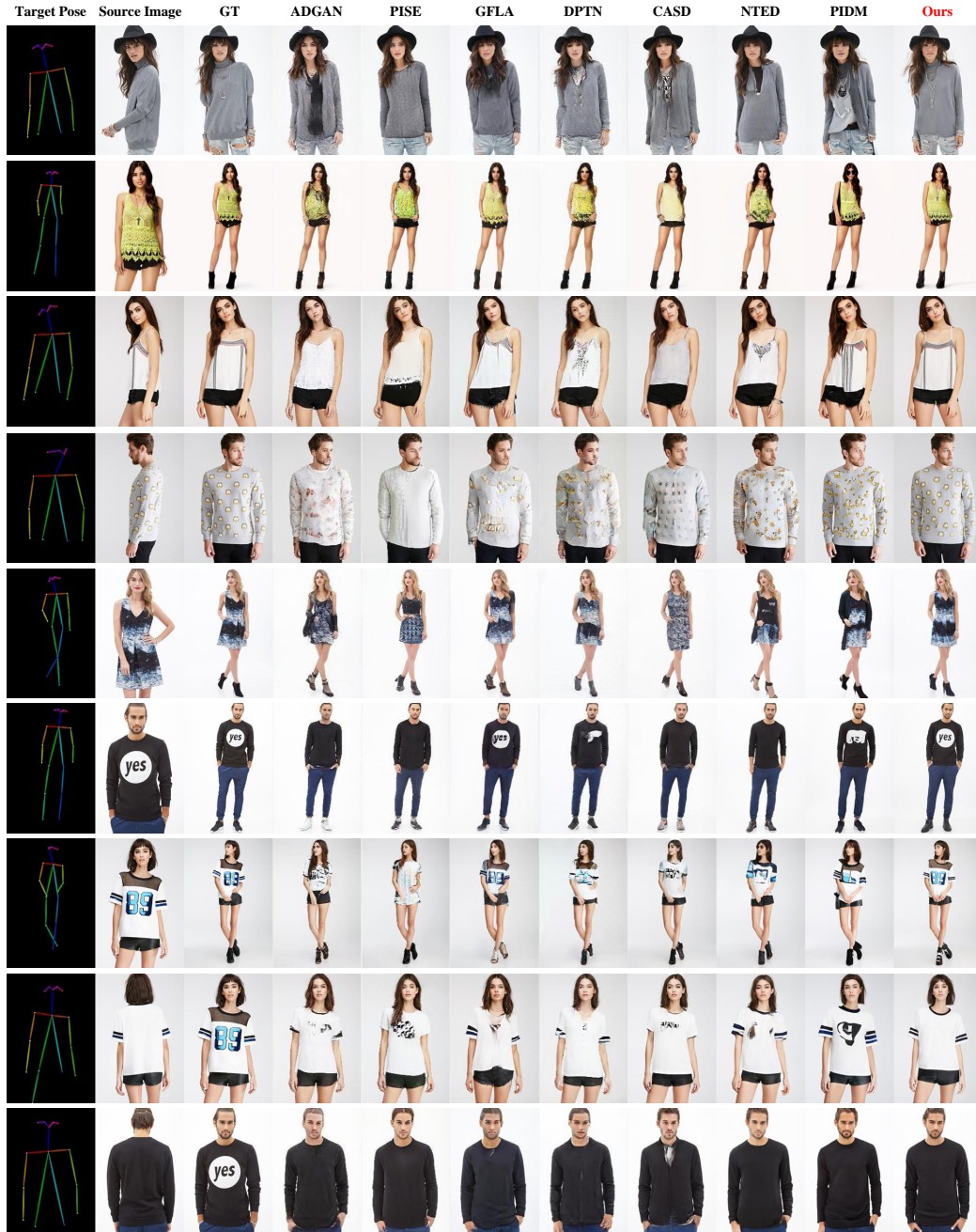

Figure 6: Qualitative comparisons with several state-of-the-art models on the DeepFashion dataset.

diffusion model. For example, when using the flow framework (i.e., CASD), PCDMs outperforms the SOTA flow-based method, CASD, even without explicitly decoupling pose and style. Although PCDMs worse FID score than PIDM (which also employs the diffusion model), we surpass it on the other two metrics, and subsequent experiments further demonstrate our method's superiority.

The comparison results on the Market-1501 are summarized in Table 1. Notably, PCDMs outshines all SOTA methods, achieving the best SSIM, LPIPS, and FID in all experiments. Specifically, compared to methods that consider fine-grained texture features, PCDMs outperforms NTED significantly in terms of both LPIPS and SSIM. While NTED explores the texture feature of the target in the source image and target pose, PCDMs extract target features by mining the global alignment relationship between pose coordinates and image appearance. Furthermore, regarding SSIM, LPIPS, and FID, PCDMs performs better than PIDM, which validated that PCDMs can accurately transfer the texture from the source image to the target pose while maintaining high consistency and realism.

**Qualitative Results.** As shown in Figure 6, we comprehensively compare our PCDMs and other state-of-the-art methods on the DeepFashion dataset. Several observations can be drawn from the results: (1) despite the minuscule size of the necklace in the source image (as seen in the first and second rows), only our proposed PCDMs and the PIDM, which also utilizes a diffusion model, can focus on it. But PCDMs can generate higher-quality images compared to the PIDM. (2) In scenarios involving extreme poses and large area occlusions (as seen in the third and fourth rows), only our method can generate images that align reasonably with the target. This can be attributed to our method's ability to capture and enhance the global features of the target image. (3) In situations with complex textures and numbers (as seen in the fifth to seventh rows), our method significantly surpasses others in preserving appearance textures, primarily due to our method's capability to refine textures and maintain consistency. (4) The last two rows present source images with invisible logo and target images with visible logo. The results indicate that PCDMs do not overfit, and our results demonstrate better visual consistency than other SOTA methods. To sum up, our method consistently produces more realistic and lifelike person images, demonstrating the advantage of our PCDMs' multi-stage progressive generation approach. See C.2 for more examples.

**User Study.** The above quantitative and qualitative comparisons underscore the substantial superiority of our proposed PCDMs in generating results. However, tasks of pose-guided person image synthesis are typically human perception-oriented. Consequently, we conducted a user study involving 30 volunteers with computer vision backgrounds.

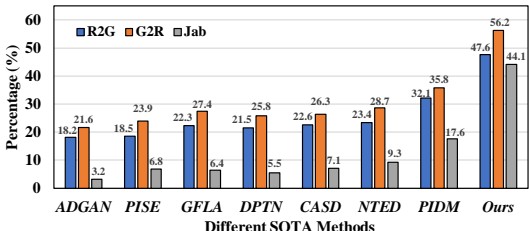

Figure 7: User study results on DeepFashion in terms of *R2G*, *G2R* and *Jab* metric. Higher values in these three metrics indicate better performance.

This study included comparisons with fundamental facts (namely, R2G and G2R) and comparisons with other methods (i.e., J2b). A higher score on these three metrics indicates superior performance. As shown in Figure 7, PCDMs exhibit commendable performance across all three baseline metrics on DeepFashion. For instance, the proportion of PCDMs images perceived as real in all instances is 56.2% (G2R), nearly 20.4% higher than the next best model. Our Jab score stands at 44.1%, suggesting a preference for our method among the participants. Please refer to C.6 of the Appendix for more detail.

## 4.2 ABLATION STUDY

To demonstrate the efficacy of each stage introduced within this paper, we have devised the following variations, all of which fall under the PCDMs framework but incorporate distinct configurations. **B1** represents using the inpainting conditional diffusion model only. **B2** denotes the simultaneous use of the prior conditional diffusion model and the inpainting conditional diffusion model without using the refining conditional diffusion model. **B3** stands for the simultaneous use of the inpainting conditional diffusion model and the refining conditional diffusion model, excluding the prior conditional diffusion model.

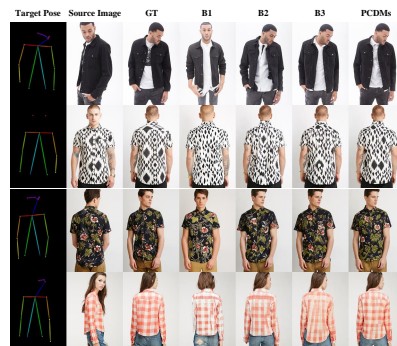

Figure 8: Qualitative ablation results. See C.3 for more examples.

Figure 8 shows the impact of each stage on the DeepFashion. B1 can generate person images that roughly conform to the target pose. However, there are severe distortions in the generated image, such as limb loss and confusion between appearance and limbs. This indicates that generating images from unaligned inputs is a highly challenging task without global features. In contrast, in B2, once the global features of the target image obtained from the prior conditional diffusion model are added, the generated person images basically match the target pose in structure. Although B2's capability to generate images with consistent appearance is somewhat limited, it has already achieved a striking resemblance to the actual image. This shows that the refining conditional diffusion model can establish a dense correspondence between the source and target images, enhancing the contextual features. In addition, from a visual perception perspective, B3 is superior to B1 and B2 regarding detail texture, while it is slightly inferior to B2 regarding pose coordination. Finally, when we use the PCDMs of the three stages in the last column, it is visually superior to B1,

B2, and B3. This indicates that when dealing with complex poses, the PCDMs of the three stages can gradually produce visually more satisfying results.

To more comprehensively validate the effectiveness of the proposed refining conditional diffusion model, we apply it to other state-of-the-art (SOTA) methods. As shown in Figure 9, the first and fourth rows denote the source image and ground truth (target image). The second and third rows present the results before and after refinement via refining the conditional diffusion model. We can observe that the refining conditional diffusion model significantly improves the results of all state-of-the-art methods. For instance, when dealing with ADGAN and PIDM, our method helps to fill in minor details, such as missing belts and hats, thereby enhancing the completeness of the generated images. For methods like GFLA, DPTN, CASD, and NTED, our model can finely process textures, maintain shape and

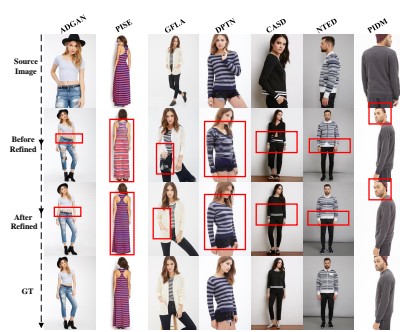

Figure 9: Effect of refining conditional diffusion model on other SOTA methods.

texture consistency, and generate more explicit and realistic images. These results indicate that the refining conditional diffusion model has universality across different state-of-the-art methods, offering potential improvements for various person image generation tasks.

### 4.3 APPLICATION

In Table 2, we further evaluate the applicability of images generated by PCDMs in a downstream task, i.e., person re-identification (Ye et al., 2021; Shen et al., 2023a). We conduct these re-identification experiments on the Market-1501 dataset, adhering to the PIDM protocol. Ini-

Table 2: Comparison with SOTA on person re-identification. *Standard* denotes not using synthesized pedestrian images.

| Methods | Percentage of real images | | | | 100% |
|---|---|---|---|---|---|
| | 20% | 40% | 60% | 80% | |
| Standard | 33.4 | 56.6 | 64.9 | 69.2 | 76.7 |
| PTN (Zhu et al., 2019) | 55.6 | 57.3 | 67.1 | 72.5 | 76.8 |
| GFLA (Ren et al., 2020) | 57.3 | 59.7 | 67.6 | 73.2 | 76.8 |
| DPTN (Zhang et al., 2022) | 58.1 | 62.6 | 69.0 | 74.2 | 77.1 |
| PIDM (Bhunia et al., 2023) | 61.3 | 64.8 | 71.6 | 75.3 | 78.4 |
| **PCDMs (Ours)** | **63.8** | **67.1** | **73.3** | **76.4** | **80.3** |

tially, we randomly select subsets of 20%, 40%, 60%, and 80% from the total real training set of the Market-1501 dataset, ensuring each identity is represented by at least one image. This selection process yields a new dataset. Following this, we employ BoT (Luo et al., 2019) as the base network and conduct baseline training with each subset of the data. We then generate synthetic images from the new dataset, randomly selecting those that share the same identity and pose. This synthesized data is then merged with the original dataset to train BoT. The Rank1 results are presented in Table 2. The results indicate that PCDMs significantly boost the re-identification performance compared to the baseline. Furthermore, when compared with state-of-the-art methods such as PTN, GFLA, DPTN, and PIDM, PCDMs consistently demonstrate superior performance in re-identification tasks.

## 5 CONCLUSION

This paper has demonstrated the significant potential of **P**rogressive **C**onditional **D**iffusion **M**odel**s** (PCDMs) in addressing the challenges of pose-guided person image synthesis through a three-stage process. In the first stage, a simple prior conditional diffusion model is designed to predict the global features of the target image by mining the global alignment relationship between pose coordinates and image appearance. The second stage establishes a dense correspondence between the source and target images using the global features from the previous stage, and an inpainting conditional diffusion model is proposed to further align and enhance the contextual features, generating a coarse-grained person image. In the final stage, a refining conditional diffusion model is proposed to utilize the coarsely generated image from the previous stage as a condition, achieving texture restoration and enhancing fine-detail consistency. The three stages of the PCDMs work progressively to generate the final high-quality and high-fidelity synthesized image. Both qualitative and quantitative results have demonstrated the consistency and photorealism of our proposed PCDMs under challenging scenarios.

**Future Work.** Our method significantly improves person image synthesis quality, while the use of two additional prior and refining models leads to increased computational resource consumption and longer inference time. Future work should explore efficient methods that provide equivalent or superior quality while reducing computational overhead and inference time.

# 6 ETHICS STATEMENT

This study introduces a novel multi-stage person image synthesis technique capable of generating new person images based on different poses and original images. However, there is a risk that malicious actors could misuse this manipulation of real photos to create false content and disseminate misinformation. This is a well-known issue faced by virtually all person image synthesis methods. Nevertheless, research has made significant progress in identifying and preventing malicious tampering. Our work will provide valuable support for research in this field and external audits, helping to balance its value against the risks posed by unrestricted open access. This will ensure that this technology can be used safely and beneficially.

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

## SUPPLEMENTARY MATERIAL

This supplementary material offers a more detailed exploration of the experiments and methodologies proposed in the main paper. Section A provides a series of symbols and definitions for enhanced comprehension. Section B delves deeper into the implementation specifics of our experiments. Section C presents additional experimental outcomes, including a broader range of qualitative comparison examples with state-of-the-art methods, an expanded set of ablation study cases, supplemented results from random sampling, and a detailed explanation of our user studies.

## A   SOME NOTATIONS AND DEFINITIONS

Table 3: Some notations and definitions.

| Notation | Definition |
|---|---|
| $x_0$ | Real image |
| $c$ | Additional condition |
| $t$ | Timestep |
| $\theta$ | Diffusion model |
| $\epsilon$ | Gaussian noise |
| $w$ | Guidance scale |
| $x_t$ | Noisy data at $t$ step |
| $x_s$ | Feature of source image |
| $p_s$ | Feature of source pose coordinate |
| $p_t$ | Feature of target pose coordinate |
| $f_{st}$ | Feature of source and target image |
| $i_{sm}$ | Feature of source and mask image |
| $p_{st}$ | Feature of source and target pose coordinate |
| $i_{ct}$ | Feature of coarse target image |

## B   IMPLEMENT DETAILS

Table 4: Hyperparameters for the PCDMs. All models trained on 8 V100 GPUs.

| Hyperparameters | Frist Stage | Second Stage | Third Stage |
|---|---|---|---|
| Diffusion Steps | 1000 | 1000 | 1000 |
| Noise Schedule | cosine | linear | linear |
| Optimizer | AdamW | AdamW | AdamW |
| Weight Decay | 0.01 | 0.01 | 0.01 |
| Batch Size | 256 | 128 | 128 |
| Iterations | 100k | 200k | 100k |
| Learning Rate | $1e-4$ | $1e-4$ | $1e-4$ |

Our experiments are conducted on 8 NVIDIA V100 GPUs. We follow the standard training strategies and hyperparameters of diffusion models. We utilize the AdamW optimizer with a consistent learning rate of $1e^{-4}$ across all stages. The probability of random dropout for condition $c$ is set at 10%. For the prior conditional diffusion model, we employ OpenCLIP ViT-H/14 [2] as the CLIP image encoder. This model's transformer consists of 20 transformer blocks, each with a width of 2,048. The model is trained for 100k iterations with a batch size of 256, using a cosine noising schedule (Nichol & Dhariwal, 2021) with 1000 timesteps. For the inpainting and refining models, we use DINOv2-G/14 [3] as the image encoder. We leverage the pretrained stable diffusion V2.1 [4], modifying the first convolution layer to accommodate additional conditions. These models are trained for 200k and 100k iterations, respectively, each with a batch size of 128, and a linear noise schedule with 1000 timesteps is applied. In the inference stage, we use the DDIM sampler with 20 steps and set the guidance scale w to 2.0 for PCDMs on all stages.

---

[2]https://github.com/mlfoundations/open_clip
[3]https://github.com/facebookresearch/dinov2
[4]https://huggingface.co/stabilityai/stable-diffusion-2-1-base

# C  ADDITIONAL RESULTS

## C.1  MORE QUALITATIVE RESULTS WITH REFINING CONDITIONAL DIFFUSION MODEL.

We show the qualitative results of PIDM combined with refining conditional diffusion model in Figure 10.

## C.2  MORE QUALITATIVE COMPARISONS FOR PCDMS

We provide additional examples for comparison with the state-of-the-art (SOTA) methods in Figure 11.

## C.3  MORE QUALITATIVE ABLATION RESULTS ON THE DEEPFASHION DATASET

We show more qualitative ablation results from our proposed PCDMs in Figure 12. **B1** signifies the exclusive use of the inpainting conditional diffusion model. **B2** represents the concurrent application of both the prior and inpainting conditional diffusion models, without incorporating the refining conditional diffusion model. **B3** denotes the combined usage of the inpainting and refining conditional diffusion models, while excluding the prior conditional diffusion model.

## C.4  RANDOM SAMPLES

We show random samples from our proposed PCDMs in Figure 13 and Figure 14. The generated results are highly stable and realistic.

## C.5  IMPACT OF PRETRAINED

We show random samples from our proposed PCDMs in Figure 15. The results show that even without pretrained, the outcomes are still remarkably realistic. This indicates that the improvement in text reconstruction is primarily due to the rationality and superiority of our framework.

## C.6  USER STUDY

In Section 4.1, we conduct a user study on the pose-guided person image synthesis task. We invite 30 volunteers with a computer science background to participate in a side-by-side view comparison of PCDMs and state-of-the-art work and to identify the authenticity of images. These samples are randomly drawn from our toolkit test set, originating from the same original image and pose. We conducted repeated tests with 50 samples for each model. The user study evaluates which model produces more realistic results through human perception. Example questions are shown in Figure 16 and Figure 17.

## C.7  VISUALIZE THE DIFFUSION PROCESS

To facilitate a better understanding of the diffusion process, we have added visualizations for the second and third stages of the diffusion process, as shown in Figure 18 and Figure 19.

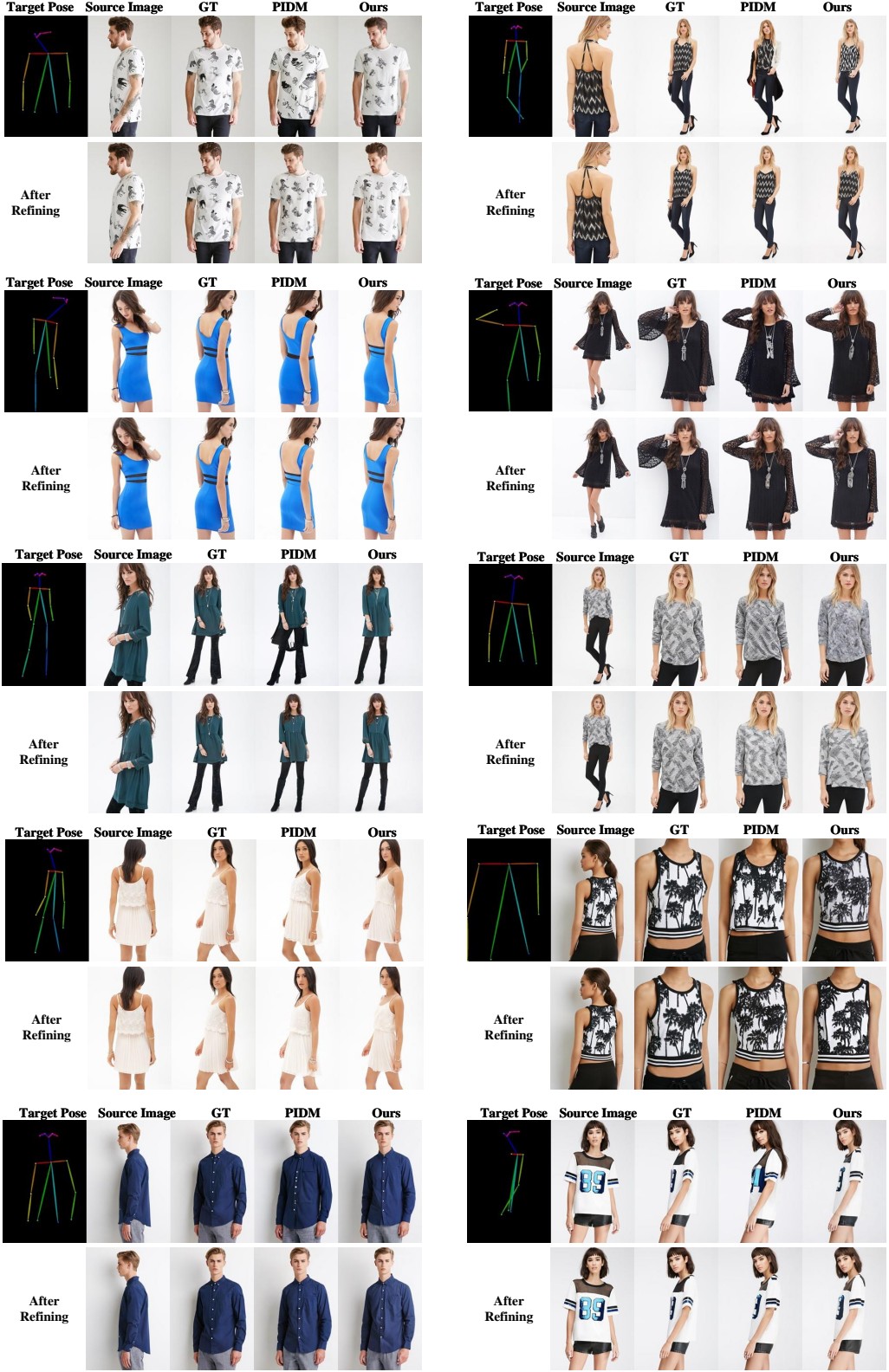

Figure 10: More examples show the refining conditional diffusion model on PIDM. It is worth noting that *ours* denotes results obtained only using the prior conditional diffusion model and inpainting conditional diffusion mode.

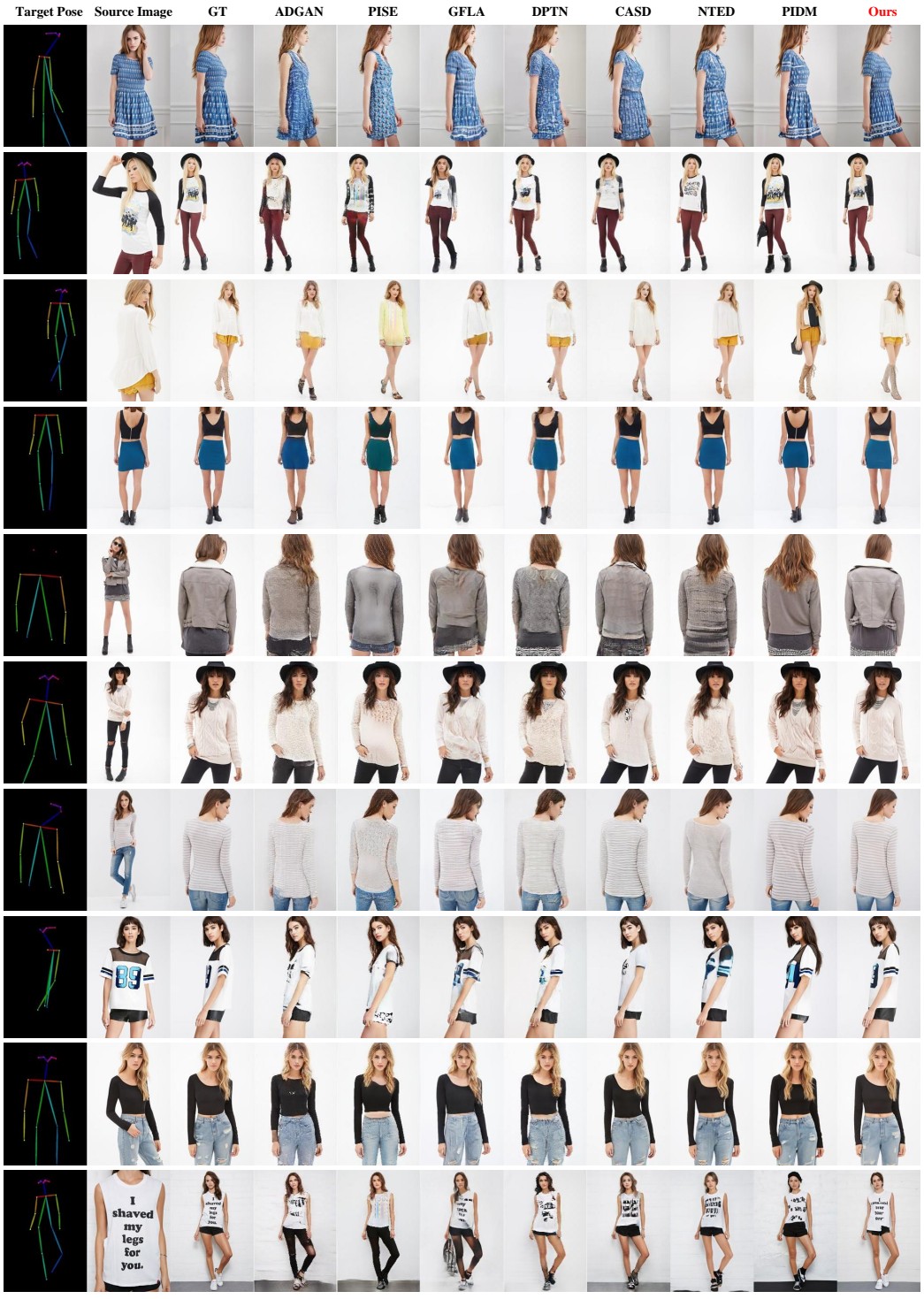

Figure 11: More qualitative comparisons between PCDMs and SOTA methods on the DeepFashion dataset.

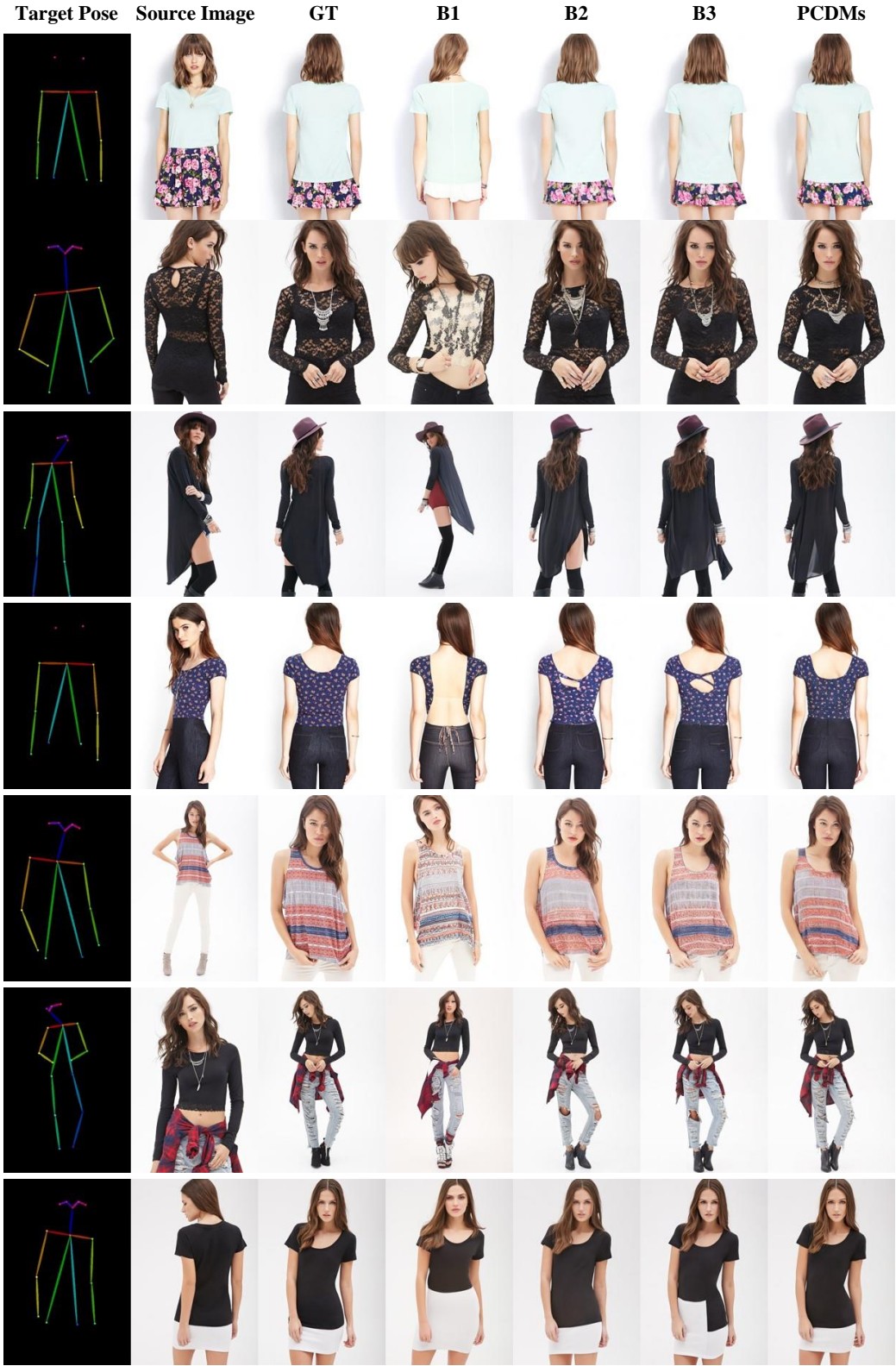

Figure 12: More qualitative ablation results on the DeepFashion dataset.

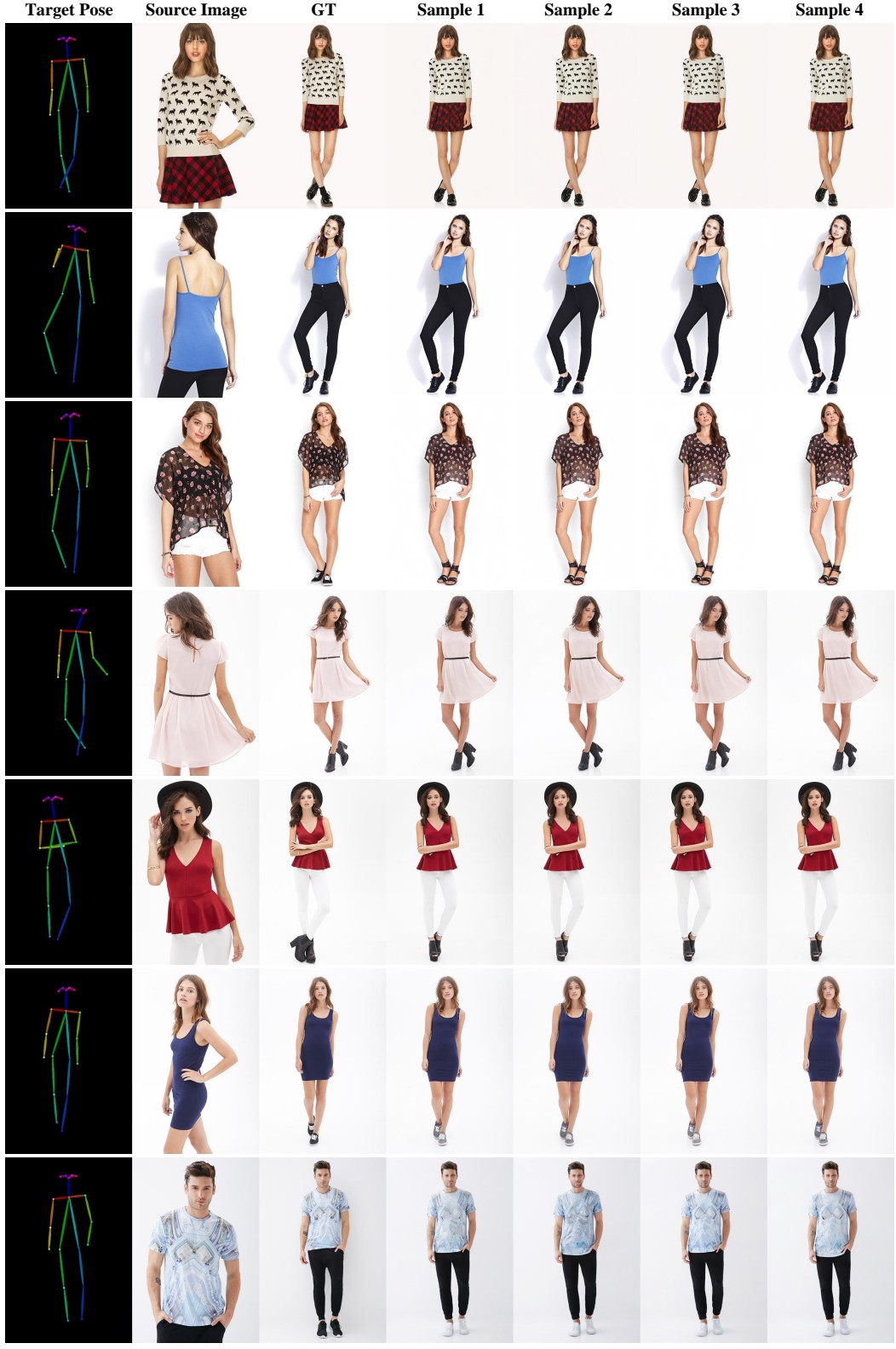

Figure 13: Random samples from DeepFashion dataset by PCDMs.

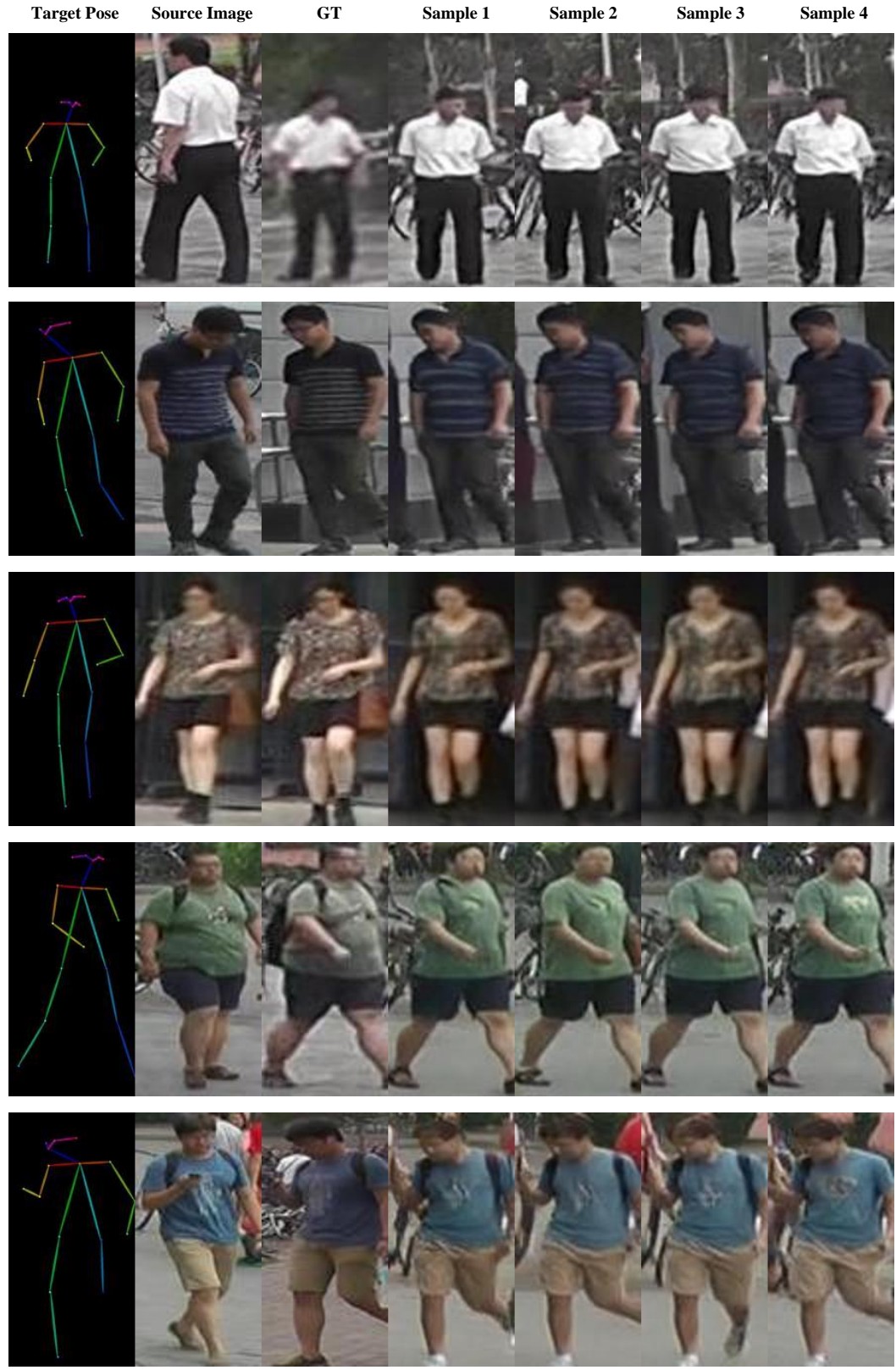

Figure 14: Random samples from Market-1501 dataset by PCDMs.

| Target Pose | Source Image | GT | *w/o* Pretrained | *w/* Pretrained |
| --- | --- | --- | --- | --- |

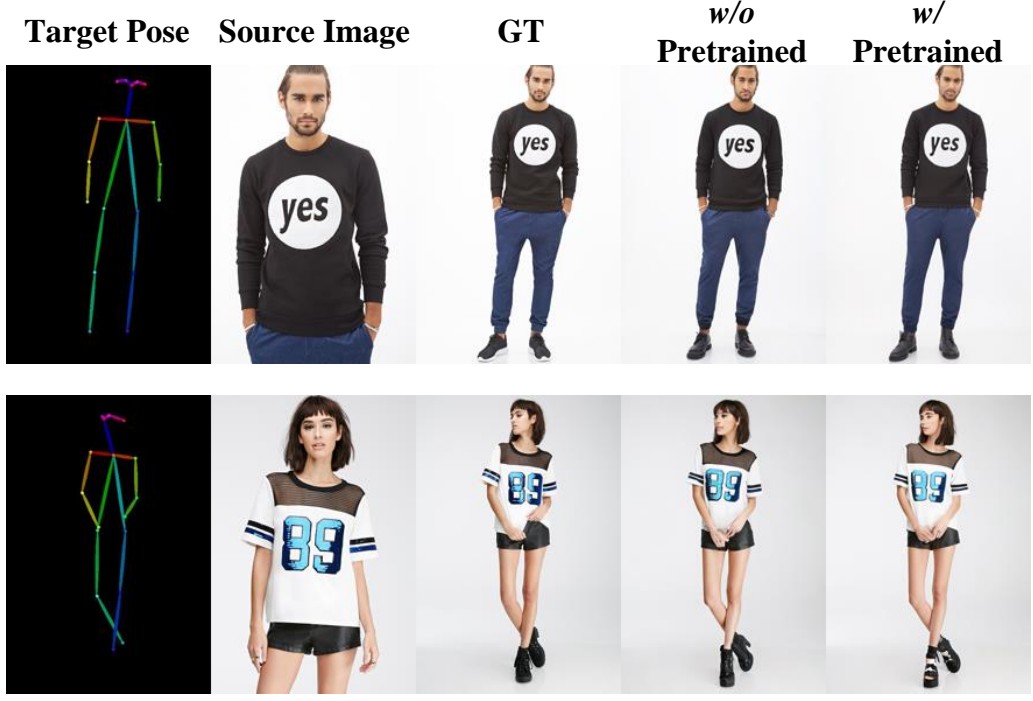

Figure 15: Compared with *w/o* pretrained weghts on PCDMs.

Please choose the most lifelike image in each group.

18.

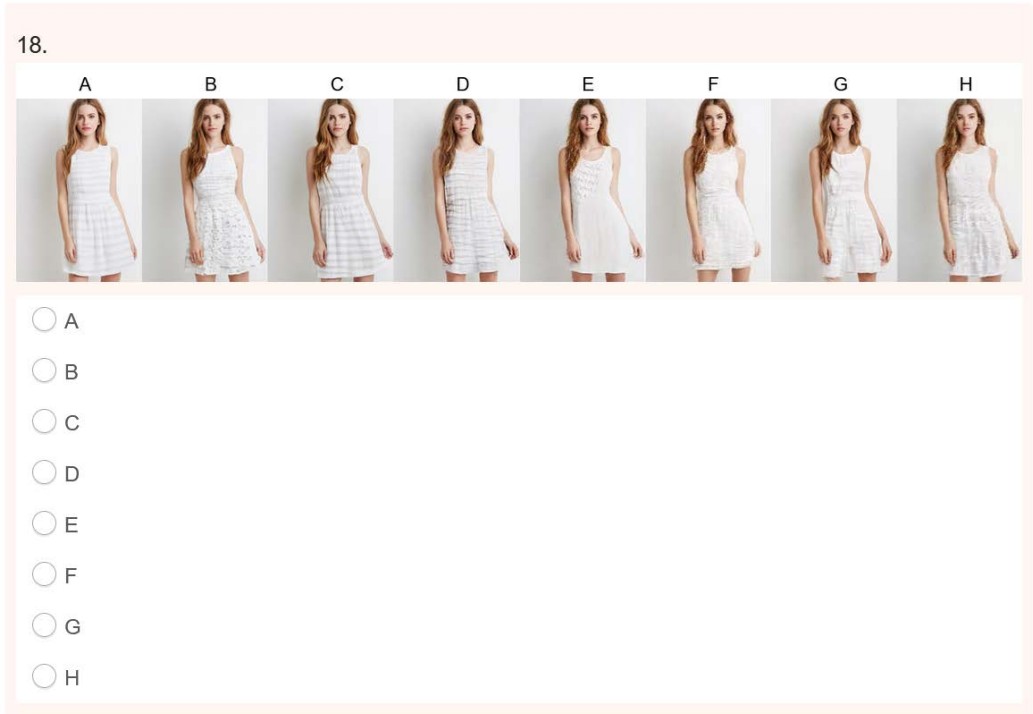

Figure 16: An example question used in our user study for pose-guided person image synthesis.

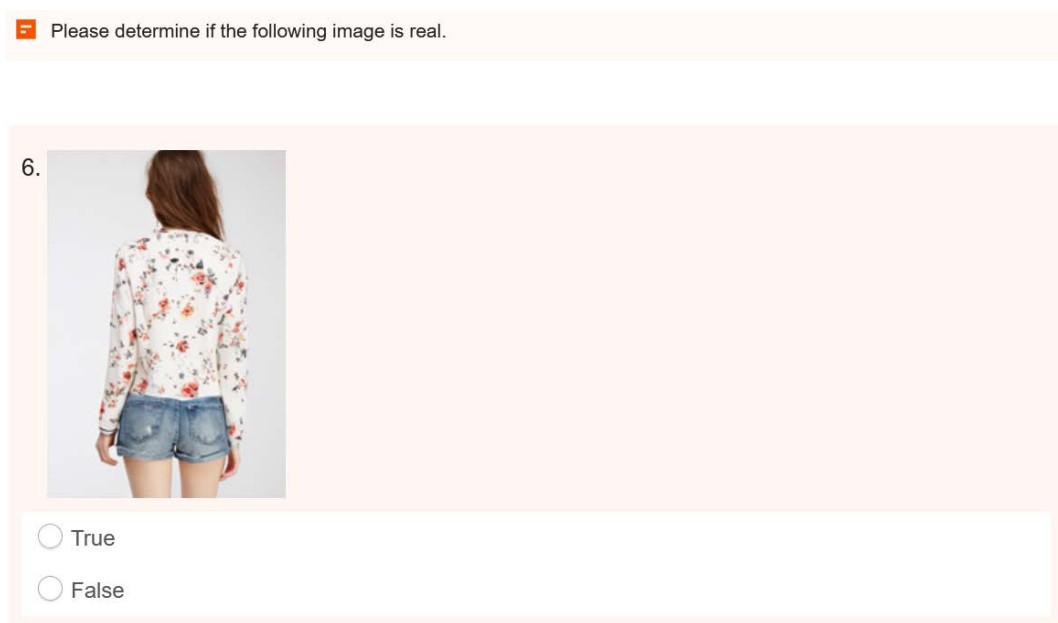

Figure 17: An example question used in our user study for pose-guided person image synthesis.

**1 - 5 Step**

**6 - 10 Step**

**11 - 15 Step**

**16 - 20 Step**

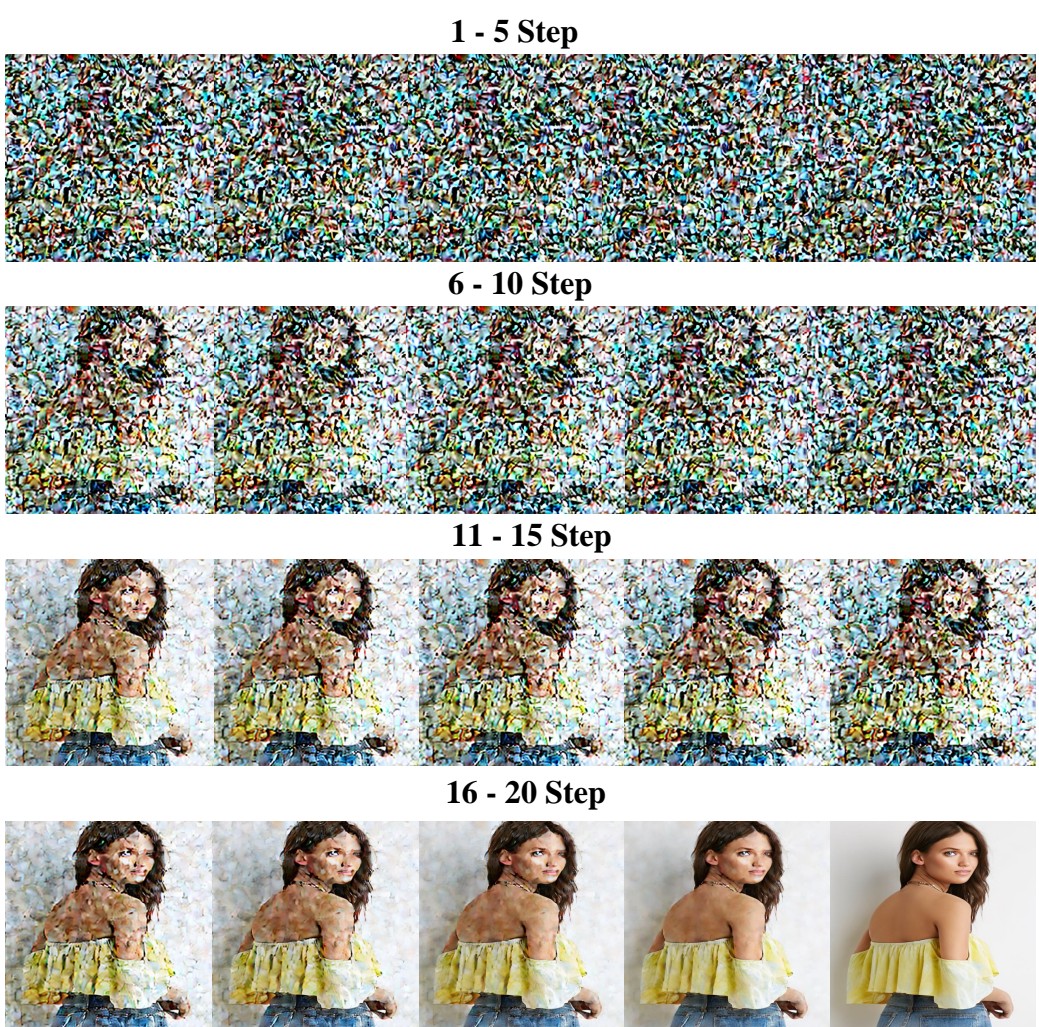

Figure 18: Visualize the diffusion process on the second stage.

**1 - 5 Step**

**6 - 10 Step**

**11 - 15 Step**

**16 - 20 Step**

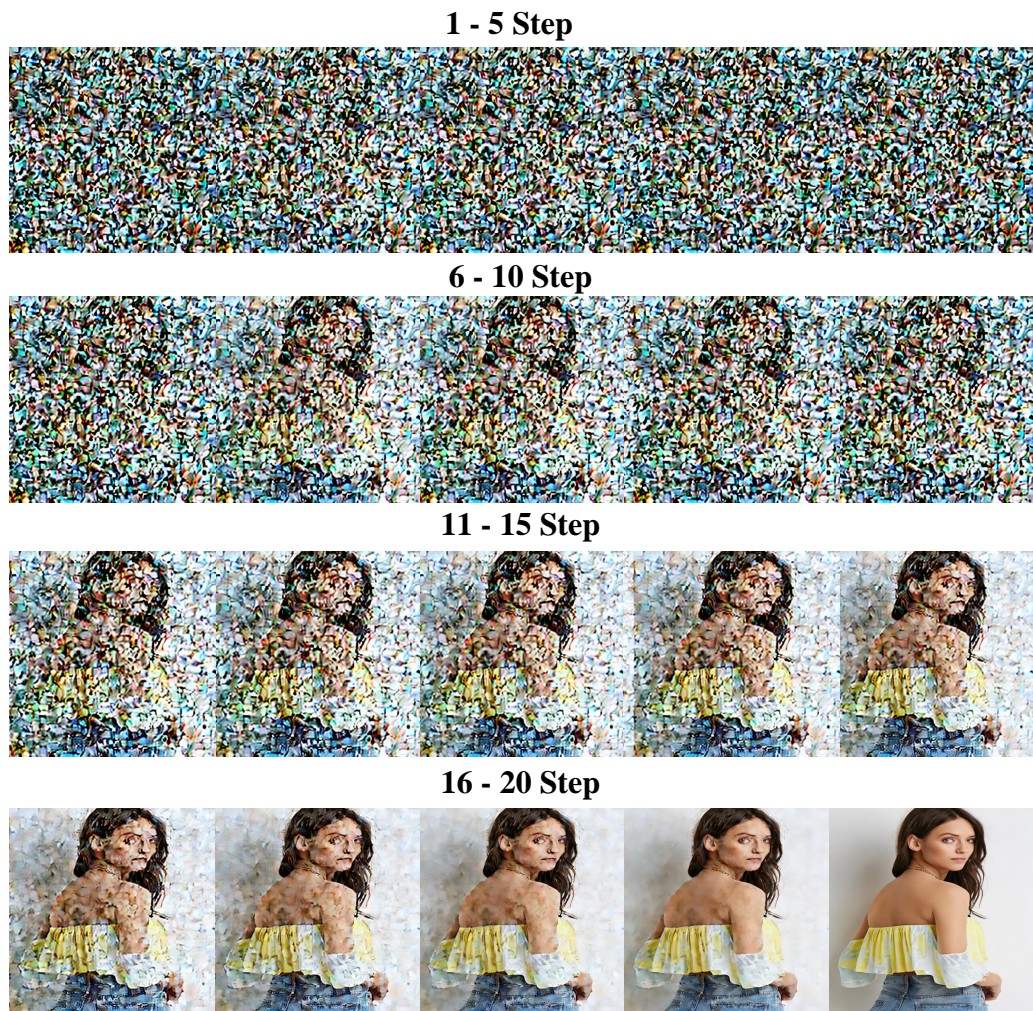

Figure 19: Visualize the diffusion process on the third stage.

