# OpenReview forum: "Advancing Pose-Guided Image Synthesis with Progressive Conditional Diffusion Models"
_ICLR.cc/2024/Conference — ICLR 2024 poster_

### Official Review · Reviewer_dqE5 · 2023-10-30

**Soundness:** 3 good
**Presentation:** 3 good
**Contribution:** 2 fair
**Rating:** 5
**Confidence:** 4

**Summary:**

The paper proposes a new pipeline based on image diffusion to tackle pose-guideed human image generation problem. The problem is decomposed into three stages, each stage requiring a separate network. At least two stages use a diffusion process, which means they involve multiple iterations, with a neural network inference step at each iteration. The first stage extracts features from the source image and source and target pose encodings. Next stage generates a coarse image, and the final stage involves refinement of the image.

The results outperform state-of-the-art in most metrics with a significant margin. The code and models will be available upon acceptance.

**Strengths:**

The paper outperforms state-of-the-art on a challenging DeepFashion dataset, as well as on the Market-1501 dataset, both in algorithmic metrics and according to a user study. User study in particular shows a big gap between the results of the proposed method and the competitive methods.

**Weaknesses:**

The paper does not clearly define its novelty. The contributions are formulated very vaguely. For example, "we propose a novel inpainting conditional diffusion model to explore the dense correspondence between source and target images" - what does "explore" really mean in this sentence? "we introduce a new refining conditional diffusion model by further using post hoc image-to-image techniques to enhance the quality and fidelity of synthesized images." - what is new in this model? Is it a typical diffusion model for image refinement, or is there something special? All in all, it is hard to say whether the theoretical/technical novelty is significant enough.

Next, the method is described quite inaccurately. Only red arrows in Fig. 2 indicate that there is some iterative process of image generation. The paper should describe in details the whole process.

**Questions:**

What is novel in the submission? I do not assume there is no novelty at all, but I'd like to see a clear formulation of the novelty.

How does the whole process of image generation look like? How many iterations are used?

x_t should go under the expectation (E) symbol in (3), because x_t is x_0 plus random noise, am I right?

"Additionally, we add an extra embedding to predict
the target global embedding." - what is the target global embedding exactly? Is it a feature vector representation of the target image?

Fig. 3 shows some noisy image labelled as ‘Target’ fed into the Image Encoder - what is that exactly?

In 3.3., “ To prevent confusion between black and white in the source and
target images,” - unclear

"the timestep embedding" - means the timestep of the diffusion model, right?

In (5), x_0 is under the expectation symbol, but not in the formula - why?

In Fig. 4, what do the noisy Source and Target images actually represent?

In general, to disuambiguate the text, I would suggest to reserve a term ‘target’ only for the dataset target image, and call a generated image aimed to resemble the target as ‘target estimate’.

In the experiments section there are no images for ‘Market-1501’ results. The results for DeepFashion show un-natural faces, and all faces look the same. This makes me think that the performance of the model for person re-identification will be limited, however the experiments show good performance on Market-1501. Would be nice to see some illustrations.

How long does it take to generate a new image using the proposed pipeline?

---

> ### Author Response · Authors · 2023-11-21
> **Thank you! Responses to Reviewer dqE5 (Part 1)**
>
> Dear Reviewer dqE5:
>
> Thank you very much for  the detailed comments. We address your concerns as follows.
>
>
> **Q1: What is novel in the submission? I do not assume there is no novelty at all, but I'd like to see a clear formulation of the novelty.**
>
> **Response:** Pose-guided image synthesis, which involves generating realistic images closely aligned with given pose and appearance information, is a highly challenging task. In existing research, most methods focus on directly generating pedestrian images that match the target pose from the source image and target pose. See: [Figure1](https://drive.google.com/file/d/1cAIz1CR0o8rh1abPlfG7p2GtxWyXiLdB/view?usp=drive_link). However, directly capturing the complex structure of spatial transformations is quite challenging, especially when it is necessary to infer obscured body areas from the given source image, which often results in noticeable artifacts. See: [https://drive.google.com/file/d/16lBLXR6wcb5KRAfW3rLVaelQ0c9cfk0r/view?usp=drive_link](https://drive.google.com/file/d/16lBLXR6wcb5KRAfW3rLVaelQ0c9cfk0r/view?usp=drive_link) .
> Then, the pose of the source image and the target image do not align, leading to misalignment in image-to-image generation. Especially when facing significant pose changes, these methods usually fail outright. See [https://drive.google.com/file/d/1ynlhikPzw6Nnp7T2x9eQa3ZMwMds-tqA/view?usp=drive_link](https://drive.google.com/file/d/1ynlhikPzw6Nnp7T2x9eQa3ZMwMds-tqA/view?usp=drive_link). Lastly, existing methods lack a mechanism for texture repair, resulting in generated results that are often not realistic enough. See [https://drive.google.com/file/d/12PECjus_hKl7KxIp4HK4eZ_tIWr1bLef/view?usp=drive_link](https://drive.google.com/file/d/12PECjus_hKl7KxIp4HK4eZ_tIWr1bLef/view?usp=drive_link). Therefore, current methods face the following problems:
>
> **1. Generation of invisible areas easily produces artifacts;**
>
> **2. Unaligned generation results fail outright when dealing with a wide range of pose changes;**
>
> **3. Lack of a mechanism for detail texture repair.**
>
> To address these issues, PCDMs decompose this task and propose a progressive conditional diffusion model. See [https://drive.google.com/file/d/1H2ssD3oUTPRu-R9GQurXjePPq_ol3Vx6/view?usp=drive_link](https://drive.google.com/file/d/1H2ssD3oUTPRu-R9GQurXjePPq_ol3Vx6/view?usp=drive_link).
> To solve problem 1, we propose a prior conditional diffusion model that directly predicts the features of the target image based on the source pose, target pose, and source image. This prediction is much simpler than directly generating the target image because the prior model only needs to focus on one task.
>
> To solve problem 2, we propose an inpainting conditional diffusion model that uses the global features predicted in the first stage to achieve alignment at the pose, feature, and image levels. The generated results will inevitably correspond densely, thus maintaining a high degree of consistency.
>
> Finally, to solve problem 3, we propose a refining conditional diffusion model that repairs the generated image in the second stage through the texture details of the source image.
>
> PCDMs incrementally bridge the gap between person images under the target and source poses through three stages, and qualitative, quantitative, and user studies demonstrate the consistency and photorealism of our proposed PCDMs compared to previous methods.
>
>
> **Q2: How does the whole process of image generation look like? How many iterations are used?**
>
> **Response:** Firstly, the prior conditional diffusion model predicts the features of the target image based on the pose of the source image, the pose of the target image, and the features of the source image. Secondly, the inpainting conditional diffusion model generates a coarse-grained target estimate based on the pose pairs of the source and target images, the source image and the mask of the target image, the features of the source image, and predicted features of the target image (which obtained by the first stage). Finally, the refining conditional diffusion model generates the fine-grained target estimate using the coarse-grained target estimate obtained from the previous stage and the source image.
> Besides, the PCDMs require 400k iterations, with the number of iterations for each stage as follows.
>
> |Stages  |Iterations |
> |:--|:--|
> |Stage1    |100k |
> |Stage2    |200k |
> |Stages    |100k |

---

> ### Author Response · Authors · 2023-11-21
> **Thank you! Responses to Reviewer dqE5 (Part 2)**
>
> **Q3: x_t should go under the expectation (E) symbol in (3), because x_t is x_0 plus random noise, am I right?**
>
> **Response:** Yes, you are right. Thank you very much for pointing out this. As $\boldsymbol{x}_t = \alpha_t\boldsymbol{x}_0+\sigma_t\boldsymbol{\epsilon}$, we should add $\boldsymbol{\epsilon}$ under the expectation (E) symbol in (3). We sincerely apologize for this typographical error and have added it in our revised version.
>
> **Q4: "Additionally, we add an extra embedding to predict the target global embedding." - what is the target global embedding exactly? Is it a feature vector representation of the target image?**
>
> **Response:** Yes, the target global embedding is a feature vetcor of target image extracted by CLIP image encoder. We will rewrite and update it in the reversion for better understanding.
>
> **Q5: Fig. 3 shows some noisy image labelled as ‘Target’ fed into the Image Encoder - what is that exactly?**
>
> **Response:** “Target” represents the target image with noise. During the training phase of the diffusion model, a noisy target image is provided to guide the model on how to recover the original image from the noise. We add noise to the global image embedding for the prior diffusion model, and the diffusion model is trained to predict the original global image embedding.
>
> **Q6: In 3.3., “ To prevent confusion between black and white in the source and target images,” – unclear**
>
> **Response:**  We thank the reviewer for pointing out this issue. In fact, we merge the source image and the masked image, with the masked image using all-zero pixels to indicate the area to be generated. If there are black pixels in the source image that match the mask, it could easily mislead the model into thinking these areas are to be generated. To tackle this issue, we introduce a single-channel marker symbol, the size of which matches the width and height of the input image (not shown in the diagram). We choose to use 0 and 1 to represent pixels that are masked and unmasked, respectively. This method aids in reducing model confusion and ensures the accurate identification of the generation area.
>
>  **Q7: "the timestep embedding" - means the timestep of the diffusion model, right?**
>
> **Response:** Yes.  The timestep embedding refers to the vector representation. For diffusion model, timestep is firstly projected into a vector, then the vector is integrated into diffusion UNet.
>
> **Q8: In (5), x_0 is under the expectation symbol, but not in the formula - why?**
>
> **Response:**  As $\boldsymbol{x}_t = \alpha_t\boldsymbol{x}_0+\sigma_t\boldsymbol{\epsilon}$, $\boldsymbol{x}_t$ is sampled based on $\boldsymbol{x}_0$, so $\boldsymbol{x}_0$ should be under the expectation symbol.
>
> **Q9: In Fig. 4, what do the noisy Source and Target images actually represent?**
>
> **Response:** In Fig. 4, we concatenate the source and target images, here the noisy Source and Target image means the concatenated image with Gaussian noise. The diffusion model is trained to predict the noise added on the concatenated image.
>
> **Q10: In general, to disuambiguate the text, I would suggest to reserve a term ‘target’ only for the dataset target image, and call a generated image aimed to resemble the target as ‘target estimate’.**
>
> **Response:** We appreciate the reviewer's attention to this. In order to improve readability, we have revised several details of our presentations and will continue to update our manuscript.

---

> ### Author Response · Authors · 2023-11-21
> **Thank you! Responses to Reviewer dqE5 (Part 3)**
>
> **Q11: In the experiments section there are no images for ‘Market-1501’ results. The results for DeepFashion show un-natural faces, and all faces look the same. This makes me think that the performance of the model for person re-identification will be limited, however the experiments show good performance on Market-1501. Would be nice to see some illustrations.**
>
> **Response:** We appreciate your kind reminder.  In fact, person re-identification primarily relies on appearance and posture for matching, due to the long shooting distance of surveillance cameras, which results in low resolution of pedestrian images and blurred facial features. Market-1501 is a standard dataset for person re-identification, which also exhibits these traits. Additionally, according to your suggestion, we have added result images from Market-1501 for a better understanding. See:  [https://drive.google.com/file/d/1Au19kw-IODWdwPbmaKlGBAHHaFxxYMEC/view?usp=drive_link](https://drive.google.com/file/d/1Au19kw-IODWdwPbmaKlGBAHHaFxxYMEC/view?usp=drive_link).
>
> **Q12: How long does it take to generate a new image using the proposed pipeline?**
>
> **Response:** Thanks for the question. PCDMs can generate a 128x64 image in 2.181 seconds, a 256x176 image in 2.288 seconds, and a 512x352 image in 3.701 seconds. Moreover, we also offer the time consumption for each stage as a reference. All the experiments were carried out on the a same V100 GPU.
>
> |Size    |PCDMs     |Stage1    |Stage2     |Stage3 |
> |:----------|:----------|:---------|:---------|:------|
> |128 x 64    |2.181    |0.414    |0.899    |0.868 |
> |256 x 176    |2.288    |0.426    |0.945    |0.917 |
> |512 x 352    |3.701    |0.458    |2.125    |1.118 |
>
> Under the same configuration, the previous sota method PIDM (which also employs the diffusion model) needs 11.746s at 512x352. Despite PCDMs being a multi-stage pipeline, PCDMs is faster than PIDM as we adopt efficient latent diffusion architecture.
>
>
> **In the end, thanks a lot for your detailed comments and thank you for helping us improve our work! We appreciate your thoughts on our work and we would be more than happy to discuss more during or after the rebuttal to explore more in this direction. Please let us know if you have any further questions. We are actively available until the end of this rebuttal period.**

---

> ### Author Response · Authors · 2023-11-23
> **Seeking Further Feedback**
>
> Dear Reviewer dqE5:
>
> Again, thank you very much for the detailed comments.
>
> In our earlier response and revised manuscript, we have conducted additional experiments and provided detailed clarifications based on your questions and concerns. As we are ending the stage of the author-reviewer discussion soon, we kindly ask you to review our revised paper and our response and consider adjusting the scores if our response has addressed all your concerns. Otherwise, please let us know if there are any other questions.
> **We would be more than happy to answer any further questions.**
>
> Best,
>
> Authors

---

### Official Review · Reviewer_m9vj · 2023-10-31

**Soundness:** 3 good
**Presentation:** 3 good
**Contribution:** 3 good
**Rating:** 6
**Confidence:** 4

**Summary:**

This paper proposes Progressive Conditional Diffusion Models (PCDMs) for pose-guided person image synthesis. Unlike existing diffusion-guided methods that directly generate an image from source image and target pose, PCDMs progressively predict the global feature, inpaint a coarse target image, and further refine it with a conditional diffusion model. Qualitative and quantitative results show that the proposed method can effectively generate images that are structurally aligned with the target pose while preserving faithful and detailed texture from the source image. It achieves the state-of-the-art performance on the DeepFashion and Market-1501 datasets in terms of both objective metrics and user study.

**Strengths:**

S1: Sensible model design
The idea of aligning source and targets progressively at the image, pose, and feature levels is sensible and shown to be effective. The design of the prior conditional diffusion model and inpainting conditional diffusion model are quite interesting and technically sound.

S2: Convincing qualitative and quantitative results
The visual results show a consistent improvement from prior methods in terms of pose/structural alignment with the target pose as well as texture details and faithfulness to the source image. Quantitatively, PCDMs produce slightly higher FID than a prior method (PIDM) but outperforms existing methods in other metrics. The user study also show a clear preference for PCDM results over other methods.

S3: Good writing
The paper is well-written and easy to follow overall.

**Weaknesses:**

W1: Model complexity and computation overhead
The proposed framework is quite complicated and involves training multiple diffusion models whose inputs/outputs depend on one another. Since it is done in a multi-stage fashion, I’m wondering if the model performance is sensitive to certain training strategies or hyper-parameter tuning. Perhaps a more detailed quantitative ablation study can better justify the model design. Also, it would be good to show the computation overhead of each stage as well as the training/inference time.

W2: Incomplete ablation study
Section 4.2 shows qualitative results of the ablation study. However, quantitative comparisons of these variations are missing. Since each progressive training stage requires additional computation overhead but the visual differences between B2, B3, and full model are quite subtle, it is essential to show more evidence to justify the effectiveness of individual components.

**Questions:**

[Questions]

* What is the overall training and inference time and how is it compared to the prior methods?
* Why use different image encoders (CLIP and DINOv2) in different stages?
* Are the MLP networks in the inpainting stage (Figure 4) and refining stage (Figure 5) the same?
* Do you use the same guidance scale w in all training stages?

[Minor suggestions]

* Figures 1 and 2 seem to have a lot of redundancy. Maybe consider consolidating them into one?
* Typo in user study paragraph 2: “J2b” -> “Jab”
* Typo in ablation study paragraph 2: “Although the personal images generated by B2 can retain the appearance of the source image is limited”

---

> ### Author Response · Authors · 2023-11-21
> **Thank you! Responses to Reviewer m9vj (Part 1)**
>
> Dear Reviewer m9vj:
>
> We sincerely appreciate your detailed suggestions and encouragements, such as "quite interesting and technically sound". We took all your suggestions into consideration and updated our manuscript.
>
> **Q1:  Model complexity and computation overhead. The proposed framework is quite complicated and involves training  multiple diffusion models whose inputs/outputs depend on one another.  Since it is done in a multi-stage fashion, I’m wondering if the model  performance is sensitive to certain training strategies or  hyper-parameter tuning. Perhaps a more detailed quantitative ablation  study can better justify the model design.**
>
> **Response:**  We appreciate the reviewer's attention to this. We follow the standard training strategies and hyperparameters of diffusion models, including elements like the optimizer and learning rate.  Furthermore, to present our hyperparameters more intuitively, we have included a table outlining the main hyperparameters across three stages. See: [https://drive.google.com/file/d/11dv6ZAYHTVaqiqHKy-k00LqsFIimn0kw/view?usp=drive_link](https://drive.google.com/file/d/11dv6ZAYHTVaqiqHKy-k00LqsFIimn0kw/view?usp=drive_link).
>
> In fact, our first and second stages are trained independently, as we directly utilize the GT features of the target image in the second stage of training and only use the predicted features from the first stage as input during inference. Moreover, the results generated in the second stage are factored into the training of the third stage.  However, as depicted in [Figure 8](https://drive.google.com/file/d/1ROWOO4gQhUKtr01dKY0zrVSibsAPcnTs/view?usp=drive_link) of manuscripts, the refining conditional diffusion model in the third stage exhibits generalizability, as it can enhance other SOTA methods.
>
>
> **Q2: Incomplete ablation study Section 4.2 shows qualitative results of the ablation study. However,  quantitative comparisons of these variations are missing. Since each  progressive training stage requires additional computation overhead but  the visual differences between B2, B3, and full model are quite subtle,  it is essential to show more evidence to justify the effectiveness of  individual components.**
>
> **Response:** We appreciate your insightful suggestions. In response to your advice, we have incorporated the quantitative ablation results from **Figure 8** into the updated version.
>
> |Settings |SSIM  (↑)   |LPIPS  (↓)   |FID  (↓) |
> |:-------|:--------|:-------|:-------|
> |B1 |0.7209     |0.1693     |8.4238 |
> |B2   |0.7357    |0.1426    |7.7815 |
> |B3 |0.7378    |0.1419    |7.6924 |
> |PCDMs  |0.7444    |0.1365    |7.4738 |
>
> The experimental findings reveal that, initially, B1 can attain highly competitive performance on the DeepFashion dataset solely by employing the inpainting conditional diffusion model. Subsequently, with the integration of the prior conditional diffusion model and refining diffusion, B2 and B3 surpass B1 in terms of SSIM, LPIPS, and FID. These outcomes underscore their significant contribution to the success of our PCDMs. We extend our gratitude once again for your insightful guidance.
>
>
> **Q3: What is the overall training and inference time and how is it compared to the prior methods? Also, it would be good to  show the computation overhead of each stage as well as the training/inference time.**
>
>
> **Response:** We appreciate your kind reminder. Based on your advice, on 256 x176 sized DeepFashion, we have added a comparative experiment on training and inference time with the previous sota method PIDM (which also employs the diffusion model). All experiments were executed on the same machine, with training utilizing 8 V100 GPUs and inference conducted on a same V100 GPU for a fair comparison. **The results indicate that our PCDMs significantly outperform in training and inference time.** This is because our model is based on latent space diffusion, while PIDM is a pixel-level diffusion model.
>
> |Methods  |Training Time (H)| Inference Time (s)|
> |:------------|:-------------|:------------|
> |PIDM    | 197.7|**9.377** |
> |PCDMs    |103.8 |**2.288**|
>
> Furthermore, according to your suggestion, we have detailed the training and inference time, as well as the memory overhead for each stage.
> |Stages | Training Memory (G) |Training Time (H)|  Testing Memory (G)| Inference Time (s)
> |:------------|:------------|:-------|:------|:-------|
> |Stage1    | 9.7 x 8 GPUs|  9.6| 9.4 x 1 GPUs| 0.426 |
> |Stage2    | 29.9 x 8 GPUs| 64.8 | 13.8 x 1 GPUs| 0.945|
> |Stage3    | 16.5 x 8 GPUs| 29.4 | 10.9 x 1 GPUs| 0.917|

---

> ### Author Response · Authors · 2023-11-21
> **Thank you! Responses to Reviewer m9vj (Part 2)**
>
> **Q4: Why use different image encoders (CLIP and DINOv2) in different stages?**
>
> **Response:**  In the first stage, we need a global embedding of the target image. We chose the CLIP image encoder as CLIP image embedding can capture rich image content and style information, which can be used to guide subsequent target image synthesis. In the next two stages, we need fine-grained features of the source image. We use DINOv2 as an image encoder since previous work [1] found that DINOv2 performs better than CLIP.
>
> [1] Xi Chen, Lianghua Huang, Yu Liu, Yujun Shen, Deli Zhao, and Hengshuang Zhao. Anydoor: Zeroshot object-level image customization. arXiv preprint arXiv:2307.09481, 2023.
>
> **Q5: Are the MLP networks in the inpainting stage (Figure 4) and refining stage (Figure 5) the same?**
>
> **Response:** In the inpainting and refining stages, we all use the same MLP network to map features, but they do not share weights and are trained independently.
>
>
> **Q6: Do you use the same guidance scale w in all training stages?**
>
> **Response:** Yes, we all use the same gudiance scale ($w$=2.0) in the three stages.
>
> ----
> **Minor suggestions**
>
> **M1: Figures 1 and 2 seem to have a lot of redundancy. Maybe consider consolidating them into one?**
>
> **Response:** We sincerely appreciate the constructive comments from the reviewers. Following your suggestions, we have integrated them into the revised version, and simplified the painting to make it more intuitive. See:  [https://drive.google.com/file/d/1cAIz1CR0o8rh1abPlfG7p2GtxWyXiLdB/view?usp=drive_link](https://drive.google.com/file/d/1cAIz1CR0o8rh1abPlfG7p2GtxWyXiLdB/view?usp=drive_link) and [https://drive.google.com/file/d/1H2ssD3oUTPRu-R9GQurXjePPq_ol3Vx6/view?usp=drive_link](https://drive.google.com/file/d/1H2ssD3oUTPRu-R9GQurXjePPq_ol3Vx6/view?usp=drive_link).
>
> **M2: Typo in user study paragraph 2: “J2b” -> “Jab”   and  Typo in ablation study paragraph 2: “Although the personal images  generated by B2 can retain the appearance of the source image is limited”**
>
> **Response:** We apologize for the error in the manuscript here and greatly appreciate the reviewer's patient reading. Furthermore, we have made every effort to check the reversion thoroughly.
>
> **Again, thank you so much for helping us improve the paper! Please let us know if you have any further questions. We are actively available until the end of this rebuttal period. Looking forward to hearing back from you!**

---

> ### Author Response · Authors · 2023-11-23
> **Seeking Further Feedback**
>
> Dear Reviewer m9vj:
>
> Again, we sincerely appreciate your detailed suggestions and encouragements, such as "Sensible model design", "Convincing qualitative and quantitative results", and "Good writing", which have greatly improved our work and inspired us to research more!
>
>
> Then, in our earlier response and revised manuscript, we have conducted additional experiments and provided detailed clarifications based on your questions and concerns.
>
>
> As we are ending the stage of the author-reviewer discussion soon, we kindly ask you to review our revised paper and our response and consider adjusting the scores if our response has addressed all your concerns. Otherwise, please let us know if there are any other questions. **We would be more than happy to answer any further questions.**
>
> Best,
>
> Authors

---

### Official Review · Reviewer_hURi · 2023-11-01

**Soundness:** 4 excellent
**Presentation:** 4 excellent
**Contribution:** 4 excellent
**Rating:** 8
**Confidence:** 3

**Summary:**

This paper proposes Progressive Conditional Diffusion Models (PCDMs) that tackles pose-guided person image synthesis with three stages: 1. predicts the global feature of target image; 2. establish dense correspondence and inpaint; 3. enhance and refine details. The authors show that the proposed method outperforms prior work.

**Strengths:**

1. The proposed method is clear and well-motivated. Studying the stage decomposition for solving a specific task with diffusion models could be a worthy contribution.
2. The evaluation is comprehensive, and the proposed method show advantages over prior works in general, with user study and clear visualization.
3. The effect of different components are studied with ablations, which further demonstrated the effectiveness of the proposed techniques and provides a better understanding of the proposed method.

**Weaknesses:**

In Table 1, the FID of the proposed method is sometimes worse than the prior work PIDM. Since both models are diffusion models, what are the potential main reasons that PIDM has a better FID, which is inconsistent to the user study / visualization that shows the proposed method is better?

[minor]
"Although PCDMs have a lower FID score than PIDM" -> "worse FID score"?

**Questions:**

In Figure 6 visualization, it seems the proposed method is much better at the reconstruction of text patterns. Is this because the proposed method utilize a pretrained stable diffusion v2? How much performance gain over prior works could be due to having a pretrained stable diffusion v2?

---

> ### Author Response · Authors · 2023-11-21
> **Thank you! Responses to Reviewer hURi**
>
> Dear Reviewer hURi:
>
> Many thanks to your strong support and your questions, which help us a lot to improve our work. We address your questions as follows.
>
> **Q1: In Table 1, the FID of the proposed method is sometimes worse than the  prior work PIDM. Since both models are diffusion models, what are the  potential main reasons that PIDM has a better FID, which is inconsistent  to the user study / visualization that shows the proposed method is better?**
>
> **Response:** We really appreciate this question, and we are quite interested in talking about it more. We want to emphasize that FID differs from SSIM and LPIPS metrics, as all methods use generated images and  training set for FID calculation. Additionally, we add an FID result between the test set and the training set. We find there appears to be a significant distribution shift between the testing set and the training set. The FID score of the testing set is approximately 7.5, which is not only very close to ours but also significantly worse than that of PIDM. From this, we infer that PIDM's learned distribution is overly fitted to the training set. Therefore, the FID score may not be the most accurate indicator of model performance in this scenario. This is also why the visualizations and user study results seem inconsistent with the FID scores.
>
>
> **Q2: [minor] "Although PCDMs have a lower FID score than PIDM" -> "worse FID score"?**
>
> **Response:** We apologize for the error in the manuscript here and greatly appreciate the reviewer's patient reading. Furthermore, we have made every effort to check the revised version thoroughly.
>
>
> **Q3: In Figure 6 visualization, it seems the proposed method is much better  at the reconstruction of text patterns. Is this because the proposed  method utilize a pretrained stable diffusion v2? How much performance  gain over prior works could be due to having a pretrained stable diffusion v2?**
>
>
> **Response:** We greatly appreciate your question. We have added a set of comparisons between using pretrained and not using pretrained at the reconstruction of text patterns.  See:  [https://drive.google.com/file/d/1VMxDBfT-vQfaVXcPeMyzKFDJbXN6lT3i/view?usp=drive_link](https://drive.google.com/file/d/1VMxDBfT-vQfaVXcPeMyzKFDJbXN6lT3i/view?usp=drive_link).
> The results show that even without pretrained, the outcomes are still remarkably realistic. This indicates that the improvement in text reconstruction is primarily due to the rationality and superiority of our framework. Additionally, we have added a set of qualitative metrics, and we found that the pretrained PCDMs show a slight improvement in these metrics.
>
> |Settings |SSIM  (↑)   |LPIPS  (↓)   |FID  (↓) |
> |:--|:------|:--------|:--------|
> |w/o pretrained |0.7428     |0.1389     |7.6946 |
> | w/ pretrained  |0.7444    |0.1365    |7.4734 |
>
> **Lastly, thank you so much for helping us improve the paper and appreciate your open discussions! Please let us know if you have any further questions. We are actively available until the end of this rebuttal period. Looking forward to hearing back from you!**

---

> > ### Author Response · Authors · 2023-11-23
> > **Seeking Further Feedback**
> >
> > Dear Reviewer hURi,
> >
> > Thank you for your support and helpful comments. We've tried our best to address your concerns, and we hope our responses make sense to you. Importantly, we much value your comments and would be happy to discuss more. **Although the author-engaged discussion phase will be over by today, if you have any additional questions or open discussions, please don't be hesitant to leave more comments. We are always available at all time, to actively address any concerns or be prepared for more discussions.**
> >
> > **Your opinions are rather important for us to improve the work!**
> >
> > Thank you!
> >
> > Sincerely,
> >
> > Authors

---

### Official Review · Reviewer_nRkN · 2023-11-04

**Soundness:** 3 good
**Presentation:** 3 good
**Contribution:** 2 fair
**Rating:** 6
**Confidence:** 5

**Summary:**

This work proposes a three-stage diffusion pipeline to perform human pose transfer in a progressive way. The first stage predicts a global alignment feature to address the pose-level alignment and the second stage further aligns and enhances the contextual features to generate a coarse result. Finally, the third further refine the details according to the source image. The experiments demonstrate the SOTA performance of the proposed method.

**Strengths:**

- The proposed method is conducted in a coarse-to-fine paradigm. The technical novelty of each component is not high. While the whole system combining three diffusion stages can achieve new SOTA performance on the human pose transfer task.

- Comprehensive experiments, ablations and user studies have been conducted to show the effectiveness of the proposed method.

- The paper is well-written and easy to follow.

**Weaknesses:**

- The qualitative ablation in Figure 8 is good to show the effectiveness of each component of the proposed method. However, the quantitive ablation is also very important. Besides, I would expect adding results of PCDMs w/o refining in Table 1.

- The refining results reported on different methods are on different samples. To better understand the refining benefits for different methods, it would be good to show the results of before-refining and after-refining on the same samples both quantitatively and qualitatively. For example, authors can add visual samples in Figure 10, and add quantitative results of PIDM + refining. I wonder if PIDM + refining can also achieve close performance.

- According to the ablation study, the features predicted by the first stage is important for pose-level alignment. I would expect more analysis of the progressive alignment procedure. For example, the authors may make some visual analysis based on the features, such as pixel correspondence or warp results similar to CoCosNet.

- In the metric section, it would be good to add references for R2G, G2R, and Jab to make the paper more self-contained.

- I would suggest adding some samples of source image with invisible logo and target image with visible logo to the main paper. Otherwise, one may be concerned about the overfitting issue.

- Since the proposed method has three diffusion stages, I wonder if its time cost will be much higher. Some analysis of the inference time could be discussed, which is important for practical applications.

- The pipeline contains stages and such a coarse-to-fine paradigm has been adopted in other works in different ways. For example, [a,b] also adopt a human prior + inpaint + refinement strategies. More discussions on the whole pipeline and related works could make the paper more comprehensive.

[a] Coordinate-based Texture Inpainting for Pose-Guided Human Image Generation, CVPR’19

[b] Unselfie: Translating Selfies to Neutral-pose Portraits in the Wild, ECCV’20

**Questions:**

- A quantitive ablation study is needed.
- The qualitative refining comparisons with PIDM + refining is necessary.
- A visual analysis of the progressive alignment procedure at each stage is needed.

**Details Of Ethics Concerns:**

The potential misuse of fake human images could be discussed.

---

> ### Author Response · Authors · 2023-11-21
> **Thank you! Responses to Reviewer nRkN (Part 1)**
>
> Dear  Reviewer nRkN:
>
> Thank you very much for your support and constructive suggestions. We address your concerns as follows.
>
> **Q1: The qualitative ablation in Figure 8 is good to show the  effectiveness of each component of the proposed method. However, the  quantitive ablation is also very important. Besides, I would expect  adding results of PCDMs w/o refining in Table 1.**
>
> **Response:** Thank you for your  thoughtful advice.  Following your suggestions, we have added the quantitative ablation results of Figure 8 to the revised version.
>
>
> |Settings | SSIM  (↑)   | LPIPS  (↓)   | FID  (↓) |
> |:---------|:-------|:-------|:----------|
> |B1 |0.7209     |0.1693     |8.4238 |
> |B2   |0.7357    |0.1426    |7.7815 |
> |B3 |0.7378    |0.1419    |7.6924 |
> |PCDMs  |0.7444    |0.1365    |7.4738 |
>
> The experimental results demonstrate that, firstly, B1 (only utilizing the inpainting conditional diffusion model) can achieve highly competitive performance on the DeepFashion dataset. Secondly, B2 (Prior + Inpainting) and B3 (Inpainting + Refining) outperform B1 on the SSIM, LPIPS, and FID. These results show they are also constructive to the success of our PCDMs.
>
> Besides, according to your advice, we have added the w/o refining results to Table 1. We appreciate your thoughtful reminder once again. See: [https://drive.google.com/file/d/1F3JJ9bWHdXCdrrHr_avV5CGqjrOTk-e2/view?usp=drive_link](https://drive.google.com/file/d/1F3JJ9bWHdXCdrrHr_avV5CGqjrOTk-e2/view?usp=drive_link).
>
> **Q2: The refining results reported on different methods are on  different samples. To better understand the refining benefits for  different methods, it would be good to show the results of  before-refining and after-refining on the same samples both  quantitatively and qualitatively. For example, authors can add visual  samples in Figure 10, and add quantitative results of PIDM + refining. I  wonder if PIDM + refining can also achieve close performance.**
>
> **Response:** We wholeheartedly agree with your perspective and have redrawn Figure 10 and updated it in the revised version for better comprehension.  See: [https://drive.google.com/file/d/1ooijhEL_1VcCvxqBKMtTQZhxuxJe4cvS/view?usp=drive_link](https://drive.google.com/file/d/1ooijhEL_1VcCvxqBKMtTQZhxuxJe4cvS/view?usp=drive_link).
> Furthermore, we have added the quantitative results of PIDM combined with refining.
>
>
> |Methods    |SSIM  (↑)    |LPIPS  (↓)     |FID (↓) |
> |:---------------|:-------|:-------|:----------|
> |PIDM     |0.7312     |0.1678    |6.3671 |
> |PIDM + Refining    |0.7397    |0.1462 |    6.7384 |
> |PCDMs     |0.7444     |0.1365    |7.4738 |
>
> The experimental results show that PIDM+Refining significantly improves visual consistency, SSIM, and LPIPS metric, although there is a slight deterioration in the FID metric. Besides, PCDMs also surpass PIDM+Refining on SSIM and LPIPS metrics.
>
>
> **Q3: According to the ablation study, the features predicted by the first stage is important for pose-level alignment. I would expect more analysis of the progressive alignment procedure. For example, the authors may make some visual analysis based on the features, such as  pixel correspondence or warp results similar to CoCosNet.**
>
> **Response:** Thanks for your advice. CoCosNet introduces an explicit correspondence module to establish a dense correspondence, thereby enabling the visualization of pixel correspondence. In our first stage, we use a transformer diffusion model to predict the global embedding of the target image and the global embedding is just a global representation, so it cannot visualize the pixel correspondence. Thank you once again for your kind suggestions.
>
> **Q4: In the metric section, it would be good to add references for R2G, G2R, and Jab to make the paper more self-contained**
>
> **Response:** We appreciate the suggestion.  Our revised version has added references for R2G, G2R, and Jab.

---

> ### Author Response · Authors · 2023-11-21
> **Thank you! Responses to Reviewer nRkN (Part 2)**
>
> **Q5: I would suggest adding some samples of source image with  invisible logo and target image with visible logo to the main paper.  Otherwise, one may be concerned about the overfitting issue.**
>
> **Response:** We appreciate this point and have added examples of source images with invisible logo and target images with visible logo in Figure 6 of the main paper.  See:  [https://drive.google.com/file/d/1R0o4rjNZjOdlduhWf4TdZPF7fmv_L3r7/view?usp=drive_link](https://drive.google.com/file/d/1R0o4rjNZjOdlduhWf4TdZPF7fmv_L3r7/view?usp=drive_link).
> The results indicate that PCDMs do not overfit, and our results demonstrate better visual consistency than other SOTA methods.
>
> **Q6: Since the proposed method has three diffusion stages, I wonder if its time cost will be much higher. Some analysis of the inference time could be discussed, which is important for practical applications.**
>
> **Response:** We thank the reviewer for pointing out this issue.  We have added an experiment comparing the inference time with PIDM. All experiments were conducted on the same V100 GPU to ensure a fair comparison.
>
> |Size    |PIDM    |PCDMs  |
> |:---------|:-----------|:------------|
> |256x176     |9.377 (s)   |**2.288 (s)** |
> |512x352    |11.746 (s)  |**3.701 (s)**|
>
> The results show that PCDMs outperform PIDM, another diffusion model, regarding inference time despite PCDMs being a multi-stage pipeline. For instance, PCDMs are over twice as fast as PIDM at a resolution of 256x176, and even three times faster at 512x352. This is because our model is based on latent space diffusion, while PIDM is a pixel-level diffusion model. Furthermore, we can consider employing techniques such as quantization and distillation to achieve more efficient inference in practical applications.
>
> **Q7: The pipeline contains stages and such a coarse-to-fine paradigm  has been adopted in other works in different ways. For example, [a,b]  also adopt a human prior + inpaint + refinement strategies. More  discussions on the whole pipeline and related works could make the paper more comprehensive.**
>
> **Response:** Thank you for your thoughtful advice. In the related work, we go into detail about these methods, and we updated them in our revised version.  See:  [https://drive.google.com/file/d/16Se4EDH_ahscbNJI-SR-GPz9cK1At4Kk/view?usp=drive_link](https://drive.google.com/file/d/16Se4EDH_ahscbNJI-SR-GPz9cK1At4Kk/view?usp=drive_link).
>
>
> **Again, thanks a lot for your detailed comments and thank you for helping us improve our work! Please let us know if you have any further questions. We are actively available until the end of this rebuttal period.**

---

> ### Comment · Reviewer_nRkN · 2023-11-22
> **Thanks for the authors' responses**
>
> I appreciate the authors’ comprehensive responses to my feedback. The additional results provided have addressed most of my concerns and clearly demonstrate the superiority of the proposed method.
>
> However, I believe that visualizing the progressive alignment procedure could significantly aid readers in understanding how the diffusion process contributes to correspondence construction. Furthermore, such an analysis could also improve the technical contribution of this work.
>
> While the technical novelty of each component may be somewhat limited, the overall system effectively advances the state-of-the-art, achieving higher performance levels with less computational cost compared to previous diffusion-based methods.
>
> Given these considerations, I maintain my initial rating.

---

> ### Author Response · Authors · 2023-11-22
> **Thank you and expect more disucssions !**
>
> Dear Reviewer nRkN,
>
> Again, we sincerely appreciate your detailed suggestions and encouragements, such as "overall system effectively advances the state-of-the-art" and "achieving higher performance levels with less computational cost", which have greatly improved our work and inspired us to research more!
>
>
> **Q:  Visualizing the progressive alignment procedure**
>
> **Response:**
> I'm concerned that there might be a misunderstanding between us regarding **progressive alignment**. Methods like CoCosNet align at the pixel level through a differentiable warping function. In contrast, the goal of our first stage is to "predict the global features of a target image" to compensate for the image's invisible areas. Simultaneously, this global feature provides alignment at the feature, pose, and image levels in the second stage (considering the left and right source and target images as a whole). **Therefore, the outputs of our three stages are feature vectors, coarse-grained images, and fine-grained images, respectively. So we cannot directly compare the three stages at the pixel level.** To address this issue, we have added an experiment where we compare the similarity of the features from all three stages with the GT.
>
> |  Stages |  Scores   |
> |:-----|:----|
> |Stage1| 0.8916|
> |Stage2|  0.9237|
> |Stage3|   0.9365|
>
> The experimental results show that the results of our three stages are increasingly close to the GT.
>
>
> **If you have any additional questions or anything you would like to discuss in more detail, please feel free to let us know (the Author/Reviewer discussion deadline of 11/22 is quickly approaching). We would be more than happy to discuss further and respond promptly.**
>
> Best,
>
> Authors

---

> > ### Comment · Reviewer_nRkN · 2023-11-23
> > **Thanks for the authors' further responses**
> >
> > I understand the difficulty of correspondence visualization in the diffusion method. The feature similarity measurement is a good tool. I would expect the intermediate features can also be used to calculate the similarity scores and further decoded into the pixel space to help visualize the diffusion process.

---

> > > ### Author Response · Authors · 2023-11-23
> > > **Thank you for your understanding and constructive comments**
> > >
> > > Dear Reviewer nRkN:
> > >
> > > We greatly appreciate the reviewer's further response. In line with the additional suggestions, we plan to compute the similarity of the intermediate features and incorporate visualizations of the diffusion process for enhanced comprehension. Given the approaching rebuttal deadline, if we are unable to update you promptly, we will include the newly released data on the final page of the appendix.
> > >
> > >
> > > **In the end, thanks a lot for your detailed comments and thank you for helping us improve our work!**
> > >
> > > Thank you!
> > >
> > > Sincerely,
> > >
> > > Authors

---

> > > ### Author Response · Authors · 2023-11-23
> > > **Thank you and updates about more results and visualization**
> > >
> > > Dear Reviewer nRkN,
> > >
> > > Thank you very much for your strong support and constructive comments!
> > >
> > > We have added and discussed the  similarity scores of intermediate features  in Appendix Sec. C.7.  See: [https://drive.google.com/file/d/18mwr3IC_ggmSd2ZBgWKmM7kJo4ZKEiVw/view?usp=drive_link](https://drive.google.com/file/d/18mwr3IC_ggmSd2ZBgWKmM7kJo4ZKEiVw/view?usp=drive_link).
> > >
> > >
> > >  We also added the visualization of the diffusion process in Appendix Sec. C.8.
> > > See: [Stage2](https://drive.google.com/file/d/1urD4BFfms4L-UXZiKQy8ftrCgqQOtB62/view?usp=drive_link) and [Stage3](https://drive.google.com/file/d/1RDjJTNI3JdEmuAltxomQLwOy7dMSWdZ8/view?usp=drive_link).
> > >
> > > **If our revised manuscript and rebuttal more closely meet your expectations for the paper, we respectfully ask you to reconsider your initial rating.**
> > >
> > > **If you have any further questions or require more information to raise your initial score, please feel free to let us know.**
> > >
> > > Thank you!
> > >
> > > Sincerely,
> > >
> > > Authors

---

> > > > ### Comment · Reviewer_nRkN · 2023-11-23
> > > > **Thanks for the authors' further responses**
> > > >
> > > > Thanks for the additional results on the visualizations which make the paper more comprehensive. While the most interesting analysis would be the intermediate features visualization in the first stage, which is the key for the reposing, since Stage2 and Stage3 mainly focus on refinement. I wonder if the features in Stage1 can also be decoded into pixel spaces for visualization. Overall, I lean to accept this work.

---

> ### Author Response · Authors · 2023-11-23
> **New promising findings**
>
> Dear Reviewer nRkN,
>
> Again，thank you very much for your strong support!
>
> Firstly, we need to establish a consensus that the first stage primarily focuses on global features. Secondly, based on your suggestions, we have reviewed relevant literature and attempted to generate images directly from features using the IP-Adapter [1]. See [https://drive.google.com/file/d/1DX3XHjlR6mWY318_S1nXHSE3HHwEPPqr/view?usp=drive_link](https://drive.google.com/file/d/1DX3XHjlR6mWY318_S1nXHSE3HHwEPPqr/view?usp=drive_link). We randomly selected several major pose transformations for testing, including from front to back, from back to front, and from front to side. The results demonstrate that our global features in the first stage can maintain consistency in terms of pose (please note the image on the back, where the pose color indicates left and right). Lastly, it should be noted that the intermediate features in the first stage are all noise-infused, hence they cannot be decoded and transformed into images. This is merely a denoising process.
>
> **Thanks for your time and contributions. Feel free to let us know if you have any further questions or concerns. Thank you very much.**
>
> **If you are satisfied with our response, please consider updating your score. If you need any clarification, please feel free to contact us.**
>
> [1] Ye, Hu, et al. "Ip-adapter: Text compatible image prompt adapter for text-to-image diffusion models." arXiv preprint arXiv:2308.06721 (2023).
>
> Thank you!
>
>
> Sincerely,
>
> Authors

---

### Author Response · Authors · 2023-11-21
**General comment**

We'd like to again thank all the reviewers for their constructive suggestions and for engaging in helpful discussions. Your comments on our work are much appreciated and will help the work be stronger. Below, we summarize the revisions we have made in response to the review:

---

**Some major edits made in the current revision**:
 - Thanks to Reviewer **nRkN** and Reviewer **m9vj**, we have added the quantitive ablation for better show the effectiveness of each component.
 - Thanks to Reviewer **nRkN** , Reviewer **m9vj**, and Reviewer **dqE5**. We have discussed the training and inference time of the model to provide a more comprehensive evaluation of PCDMs.
- In response to the request of Reviewer **hURi**, we have delved deeper into the reasons why PCDMs underperform compared to PIDM on the FID metric.
- Additionally, we have included more samples (see Figures 6 and 10) to demonstrate the generated results of our PCDM and PIDM. These results alleviate concerns about overfitting and validate the effectiveness of the Refining Stage. Thanks to Reviwer **nRkN** and Reviewer **m9vj**.
- Thanks to the suggestions from Reviewer **nRkN**,  visualization results on first stage with global features show promising new findings.
- We have corrected all grammatical errors and typos, and have expanded our discussion with additional references. We extend our gratitude to all the reviewers (**nRkN** , **hURi**, **m9vj**, and  **dqE5**).
- We appreciate the constructive suggestions from Reviewer **dqE5** and Reviewer **m9vj**. In response, we have added a case study on Market-1501 and provided more details about the implementation.

**We thank you for your comments and responses. We did our best to address the concerns raised by reviewers, and we appreciate these improvements could be considered. We will also add these results and findings to our manuscript.**

---

### Meta-Review · Area_Chair_bxNk · 2023-12-07

**Metareview:**

This work proposes a three stage technique for human pose guided image synthesis. Reviewers appreciated the well-written paper with good results. Multiple reviewers raised several clarity questions as well as expressed concerns over missing ablation experiments. Authors addressed several of these issues in their author responses. It is felt that the overall contributions and good result quality outweigh the weaknesses in novelty. The reviewers did raise some valuable concerns that should be addressed in the final camera-ready version of the paper, which include adding the relevant rebuttal discussions and revisions in the main paper. The authors are encouraged to make the necessary changes to the best of their ability.

**Justification For Why Not Higher Score:**

Multiple reviewers complain about overly complex system and also some missing ablations (some of which are addressed in the author responses)

**Justification For Why Not Lower Score:**

All the reviewers except one are leaning towards acceptance. One reviewer that gave 5-score mostly raised several clarity questions and did not participate in the discussion.

---

### Decision · Program_Chairs · 2024-01-16

Accept (poster)